# Convergent evolution of small molecule pheromones in *Pristionchus* nematodes

**Chuanfu Dong[1], Cameron J Weadick[2†], Vincent Truffault[3†], Ralf J Sommer[1]\***

[1]Department for Integrative Evolutionary Biology, Max Planck Institute for Developmental Biology, Tübingen, Germany; [2]Department of Biosciences, University of Exeter, Exeter, United Kingdom; [3]Max Planck Institute for Developmental Biology, Tübingen, Germany

**Abstract** The small molecules that mediate chemical communication between nematodes—so-called 'nematode-derived-modular-metabolites' (NDMMs)—are of major interest because of their ability to regulate development, behavior, and life-history. *Pristionchus pacificus* nematodes produce an impressive diversity of structurally complex NDMMs, some of which act as primer pheromones that are capable of triggering irreversible developmental switches. Many of these NDMMs have only ever been found in *P. pacificus* but no attempts have been made to study their evolution by profiling closely related species. This study brings a comparative perspective to the biochemical study of NDMMs through the systematic MS/MS- and NMR-based analysis of exo-metabolomes from over 30 *Pristionchus* species. We identified 36 novel compounds and found evidence for the convergent evolution of complex NDMMs in separate branches of the *Pristionchus* phylogeny. Our results demonstrate that biochemical innovation is a recurrent process in *Pristionchus* nematodes, a pattern that is probably typical across the animal kingdom.

**\*For correspondence:**
ralf.sommer@tuebingen.mpg.de

†These authors contributed equally to this work

**Competing interests:** The authors declare that no competing interests exist.

## Introduction

Pheromonal signaling appears to be the most ancient and widespread form of communication used in the animal kingdom (*Agosta, 1992*; *Krieger and Breer, 1999*). Deciphering the chemical components used for pheromonal communication is key to understanding why animals release information into their social environment, and to determining how we might exploit these signaling systems for management purposes. A major challenge to achieving these aims is the simple fact that pheromones are exceptionally diverse. In terms of biochemical structure, any molecule that can be released and subsequently detected can act as a pheromone. Moreover, animals often release complex blends of compounds, and they do so in context-dependent ways that are sensitive to sex, age, and condition (*Wyatt, 2003*). Finally, natural selection may shape these pheromonal signals to serve a wide variety of biological tasks, including mate detection, territorial defense, and quorum sensing (*Agosta, 1992*; *Schneider, 1992*; *Hansson, 1995*; *Rutherford and Bassler, 2012*). Consequently, pheromonal communication systems represent promising systems for exploring the evolution of diversity at the interface of biochemistry, molecular physiology, and behavioral ecology (*Symonds and Elgar, 2008*; *Missbach et al., 2014*). However, not all animals are equally suited to such integrative research—effectively studying the how and why of pheromonal communication requires systems that are both experimentally tractable and amenable to unbiased biochemical characterization.

The nematodes *Caenorhabditis elegans* and *Pristionchus pacificus* have emerged as powerful systems for studying the biochemistry and genetics of pheromonal signaling (*Ludwig and Schroeder, 2013*; *Sommer, 2015*). Nematodes have long been known to communicate via pheromones, producing both releaser pheromones (those that trigger short-term behavioral responses, for example mate attraction) and primer pheromones (those that trigger long-term developmental responses,

such as the production of quiescent dauer larvae) (*Bone and Shorey, 1978*; *Golden and Riddle, 1982*). Subsequent efforts to characterize these pheromonal signals biochemically revealed that nematodes release and respond to a diverse suite of ascarosides. Simple ascarosides consist of a 3,6-dideoxy-L-ascarylose sugar coupled to a fatty-acid-derived side chain (*Jeong et al., 2005*; *Butcher et al., 2007*). From this basic framework, nematodes produce hundreds of distinct molecules that are collectively referred to as nematode-derived modular metabolites (NDMMs). This structural diversity of NDMMs results from variation in the sugar (some NDMMs replace ascarylose with the related sugars paratose or L-3,6-dideoxy-lyxo-hexose), the fatty acid (length, saturation, and attachment position can all vary), the presence or absence of additional moieties (including both amino acid and nucleoside-derived moieties), and whether the NDMM exists as a monomer or a dimer (*Bose, 2012*; *von Reuss et al., 2012*; *Ludewig and Schroeder, 2013*; *Schroeder, 2015*; *von Reuss and Schroeder, 2015*; *Butcher, 2017a*; *Butcher, 2017b*; *von Reuss, 2018*; *Artyukhin et al., 2018*; *Butcher, 2019*; *Bergame et al., 2019*).

Several NDMMs have been confirmed to influence aspects of behavior (e.g., aggregation behavior in *C. elegans*) and development (e.g., control of the mouth-form polyphenism in *P. pacificus*) (*Srinivasan et al., 2012*; *Bose, 2012*). Moreover, genetic studies have begun to shed light on both the biochemical pathways responsible for NDMM production and the neurobiological and developmental pathways that underlie receiver response (*Butcher et al., 2009*; *von Reuss et al., 2012*; *Zhang et al., 2015*; *Leighton and Sternberg, 2016*; *Zhang et al., 2016*; *Panda et al., 2017*; *Falcke et al., 2018*; *Zhang et al., 2018*; *Zhou et al., 2018*; *McGrath and Ruvinsky, 2019*). However, we currently know very little about the evolution of NDMMs. Targeted screening for known compounds confirmed that NDMM production is widespread across the nematode phylum (*Choe et al., 2012*), and earlier work identified long-chain ascarosides in the eggshells of parasitic *Ascaris* nematodes (*Bartley et al., 1996*), but there remains a need for unbiased biochemical screens that speak to the diversity of NDMMs present both within and across species. This is in strong contrast to past research on plants and insects, where small molecules involved in signaling have been surveyed broadly and shown to undergo rapid evolutionary changes (*Symonds and Elgar, 2008*; *Engl et al., 2018*; *Lombe et al., 2019*). With densely sampled comparative data in hand, it should become possible to address key questions about NDMM evolution, such as does phylogeny reliably predict NDMM production, and how repeatable is biochemical innovation?

*Pristionchus* nematodes are well suited for phylogeny-informed research on NDMM diversity. *P. pacificus* is a free-living soil nematode that is reliably found in association with scarab beetles (*Herrmann et al., 2006a*; *Herrmann et al., 2006b*; *Herrmann et al., 2007*), and that has been developed as a model system in developmental genetics and comparative evolutionary biology. Many studies in this species focused on two examples of phenotypic plasticity, and these efforts revealed that *P. pacificus* and *C. elegans* produce and use NDMMs in distinct ways. The first example involves the development of the buccal cavity, with individuals adopting either the stenostomatous (St) mouth form with a single tooth, resulting in strict bacterial feeding, or the alternative eurystomatous (Eu) form with two teeth, which allows for predation on other nematodes (*Bento et al., 2010*; *Bumbarger et al., 2013*; *Sieriebriennikov et al., 2017*; *Sieriebriennikov and Sommer, 2018*). This behavioral-morphological polyphenism is broadly conserved within the *Pristionchus* genus and the larger Diplogastridae family (*Susoy et al., 2015*), but is not found in non-Diplogastrids such as *C. elegans*. Second, *P. pacificus* worms, similar to *C. elegans* worms, can enter an arrested and long-lived dauer stage in response to harsh environmental conditions (*Ogawa et al., 2009*). Both mouth-form plasticity and dauer induction are sensitive to population density in *P. pacificus*, with NDMMs serving as chemical indicators of competitive environments in which predation or diapause represent adaptive survival strategies (*Bento et al., 2010*; *Werner et al., 2018a*). However, completely different NDMMs are involved in the regulation of these two plastic traits (*Bose, 2012*; *Sommer and Mayer, 2015*), and *P. pacificus* and *C. elegans* employ different NDMMs as molecular triggers for dauer entry (*Bose, 2012*; *Schroeder, 2015*). Together, these studies demonstrated evidence for divergence of NDMM signaling between species and across developmental pathways.

Biochemical analysis of the *P. pacificus* exo-metabolome showed that this species produces a surprising diversity of complex NDMMs that are not found in *C. elegans*, including the dimeric NDMM dasc#1 and several highly modular compounds that are characterized by the presence of complex sidechains that incorporate intermediates from different primary metabolic pathways (the UBAS,

PASC, and NPAR NDMMs) (*Figure 1*; *Bose, 2012*; *Yim et al., 2015*; *Artyukhin et al., 2018*; *Falcke et al., 2018*). However, the common ancestor of *P. pacificus* and *C. elegans* existed around 100 million years ago (*Prabh et al., 2018*; *Werner et al., 2018b*; *Hong et al., 2019*) and comparing just these two species can provide only limited insights into patterns of small molecule evolution in nematodes. By contrast, comparative analysis involving multiple species across a shallower phylogenetic scale has the potential to shed light on the origins of complex NDMMs and the divergence of pheromone profiles. The availability of many culturable species and knowledge about phylogenetic relationships are key prerequisites for such studies, both of which are met in *Pristionchus*. Many Diplogastrid species, covering multiple genera, can be cultured in the laboratory, thereby allowing for fine-scale comparative analysis (*Susoy et al., 2015*). This diversity includes more than 40 *Pristionchus* species that have been isolated from world-wide samplings of scarab beetles and other invertebrates (*Kanzaki et al., 2012a*; *Kanzaki et al., 2012b*; *Kanzaki et al., 2012c*; *Yoshida et al., 2018*). These *Pristionchus* species fall into four major clades—the *pacificus*, *maupasi*, *entomophagus*, and *triformis* clades—plus an outgroup (*Figure 1—figure supplement 1*), nearly all of which are

**Figure 1.** Known NDMMs from *P. pacificus*. (A) Simple ascarosides, and the paratoside part#9 [par-C5]. (B) Modular NDMMs (DASC, NPAR, PASC, and UBAS chemicals) from *P. pacificus*.

The online version of this article includes the following figure supplement(s) for figure 1:

**Figure supplement 1.** Phylogeny of the genus *Pristionchus* adapted from *Rödelsperger et al., 2018*.

culturable under laboratory conditions. Within these clades, most species show geographically restricted distribution patterns that are often correlated with major continents. For example, species of the *pacificus*-clade are largely restricted to Asia. Although the majority of *Pristionchus* species are gonochoristic (comprised of males and females), self-fertilizing hermaphroditism has evolved at least six times independently within the genus, and the model species *P. pacificus* reproduces in this way (*Rödelsperger et al., 2018*; *Weadick and Sommer, 2016a*).

Here, we bring the comparative method of evolutionary biology (*Harvey and Pagel, 1991*) to the chemical analysis of NDMMs to obtain insight into their evolutionary dynamics. Exo-metabolomes derived from 32 *Pristionchus* species and six non-*Pristionchus* species from different Diplogastridae genera were prepared for a systematic analysis using HPLC-MS/MS and NMR. Through these efforts, we uncovered an unexpected structural diversity of simple and complex NDMMs, including 16 novel DASC chemicals, 12 novel UBAS chemicals, and a new family of UPAS chemicals. By analyzing these results in the light of known *Pristionchus* relationships, we found evidence that NDMM production is strongly correlated with phylogeny, with several structurally distinct NDMMs showing clade-specific distributions, and with more closely related species tending to possess more similar NDMM compositional profiles. That said, our analyses also revealed the convergent evolution of highly modular NDMMs in distantly related branches of the phylogeny, indicating that biochemical innovation is a repeatable process in *Pristionchus* nematodes. We speculate that such patterns are probably typical of pheromonal small molecules across the animal kingdom.

## Results

### Uncovering the chemical diversity of NDMMs in *Pristionchus* nematodes

*P. pacificus* produces a diverse set of NDMMs that can be classified into five major groups: simple, dimeric (DASC), NPAR, PASC, and UBAS (*Figure 1*). Of these, only a few of the simple ascarosides (i.e. ascr#9 [asc-C5], ascr#12 [asc-C6], and ascr#1 [asc-C7]) have also been identified in *C. elegans* or other nematodes. However, because of limited sampling, it is not currently possible to say whether this reflects true species-specificity, or whether the various complex compounds are shared broadly across species. Obtaining a historical and functional understanding of NDMM evolutionary innovation therefore requires a comparative perspective. To investigate the dynamics of NDMM production in *Pristionchus* nematodes, we selected 32 culturable species (*Figure 1—figure supplement 1*), including both hermaphroditic and gonochoristic species from all four major phylogeographic clades (*Figure 2*), and produced exo-metabolomes for comparative LC-MS/MS analysis (see 'Materials and methods'). Given our sample collection and biochemical analysis protocols, we employed an ion intensity threshold of $1.0 \times 10^3$: samples detected at levels above this value could be reliably quantified whereas samples detected at levels below this value were considered to be present only at trace levels.

The simple NDMMs, which often act as biosynthetic precursors for more complex compounds, consist of a single sugar coupled to a fatty acid side chain. Consistent with previous studies, we identified four simple NDMMs in *P. pacificus*: the ascarosides ascr#9, ascr#12, and ascr#1, which form a homologous series with fatty acid side chains of differing length, and the paratoside part#9, which is similar to ascr#9 but which uses paratose as the sugar moiety in place of ascarylose. We found six additional simple NDMMs when examining other *Pristionchus* species: ascr#5 [asc-ωC3], ascr#11 [asc-C4], ascr#7 [asc-ΔC7], ascr#14 [asc-C8], ascr#10 [asc-C9] and ascr#18 [asc-C11] (*Figures 2* and *3A*; *Figure 3—figure supplements 1* and *2*; *Supplementary file 3—Table 1*). Most of these are similar to ascr#9, ascr#12, and ascr#1, possessing saturated fatty acids attached via the ω−1 carbon, but ascr#5 and ascr#7 are structurally distinct: ascr#5 possesses a C3 fatty acid side chain that attaches to the sugar via the terminal ω carbon (*Butcher et al., 2008*), whereas ascr#7 possesses an unsaturated C7 side chain (*Pungaliya et al., 2009*). ascr#1, ascr#9 and ascr#12 were produced by all species, consistent with an ancient origin of these simple NDMMs (*Figure 2* and *Figure 2—source data 1*; *Supplementary file 1a—Figures 1–9*). By contrast, a few NDMMs, most notably ascr#7 (*Figure 3—figure supplement 3*; *Supplementary file 1a—Figures 10–11*) and part#9, were restricted to specific clades; ancestral reconstruction analyses suggest a single origin for ascr#7 along the branch that defines the entomophagus-clade, and two origins for part#9 within the pacificus-clade

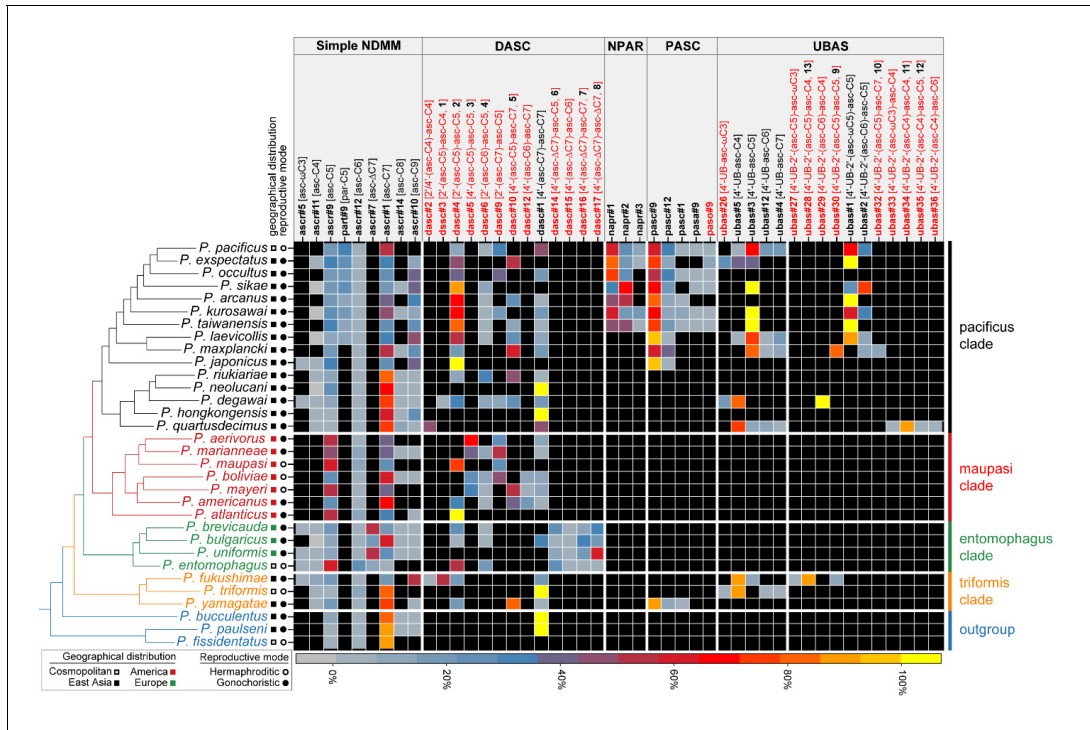

**Figure 2.** Comparative analysis of the exo-metabolomes of 32 *Pristionchus* nematodes revealed evolutionary diversity of NDMMs. Three biological replicates for each species were grown in 100 ml S-medium for comparative LC-MS/MS analysis (see 'Materials and methods'). Simple NDMMs and complex NDMMs including DASC, NPAR, PASC, and UBAS chemicals (***Supplementary file 3—Tables 1–3***) are summarized; compounds found at trace levels (including the newly described UPAS class) are not included. Values were standardized within classes to help to visualize variation in relative production levels within rare NDMM classes, as overall production levels were dominated by simple NDMMs (see ***Figure 2—source data 1***). NDMM abundances were standardized by dividing mean peak area for each NDMM by the total for the respective NDMM class then expressing the abundance as a percentage for each species; for clarity, monomeric and dimeric UBAS compounds were standardized separately. Black cells represent the apparent absence of the chemical. Black and red compound labels represent known and novel NDMMs, respectively.

The online version of this article includes the following source data for figure 2:

**Source data 1.** Quantification of NDMMs in 32 *Pristionchus* nematodes.

(***Supplementary file 4—Figures 1–2***). The remaining simple NDMMs were phylogenetically patchy, possibly indicating repeated silencing and reactivation of conserved biochemical pathways. For example, ascr#10, a known attractant of *C. elegans* hermaphrodites (***Izrayelit et al., 2012***), was absent from 6 of 7 hermaphroditic species but present in 23 of 25 gonochoristic species, and was often produced by males at particularly high levels (***Supplementary file 1a—Figures 12–24***), suggesting a broadly conserved role as a sex pheromone in gonochoristic species (***Figures 2*** and ***3B***; ***Figure 3—figure supplement 4A***). The lack of ascr#10 in hermaphroditic species might simply reflect the culture conditions (in which males are generally rare as the result of self-fertilization), but we note that ascr#10 was not detected in separate assays focused exclusively on *P. pacificus* males (***Figure 3C*** and ***Figure 3—figure supplement 4B***), consistent with a total loss of ascr#10 biosynthesis in at least one hermaphroditic species. Quantitative variation in simple NDMM levels was also apparent; depending on species, the most abundant compound was either ascr#1, ascr#9, ascr#10, or ascr#7. Moreover, a food limitation (starvation) treatment altered NDMM levels in *P. pacificus* hermaphrodites, indicating that NDMM production rates are plastic (***Figure 3C*** and ***Figure 3—figure supplement 4B***). Combined, these results demonstrate high levels of species-specificity within the simple NDMM class and indicate that NDMM production by *Pristionchus* correlates with phylogeny in some respects but is also influenced by sex, the mode of reproduction, and the environment. Similar patterns emerge from comparative study of complex NDMMs, albeit with more pronounced evidence for lineage-specificity and clear signatures of convergent evolution, as shown below.

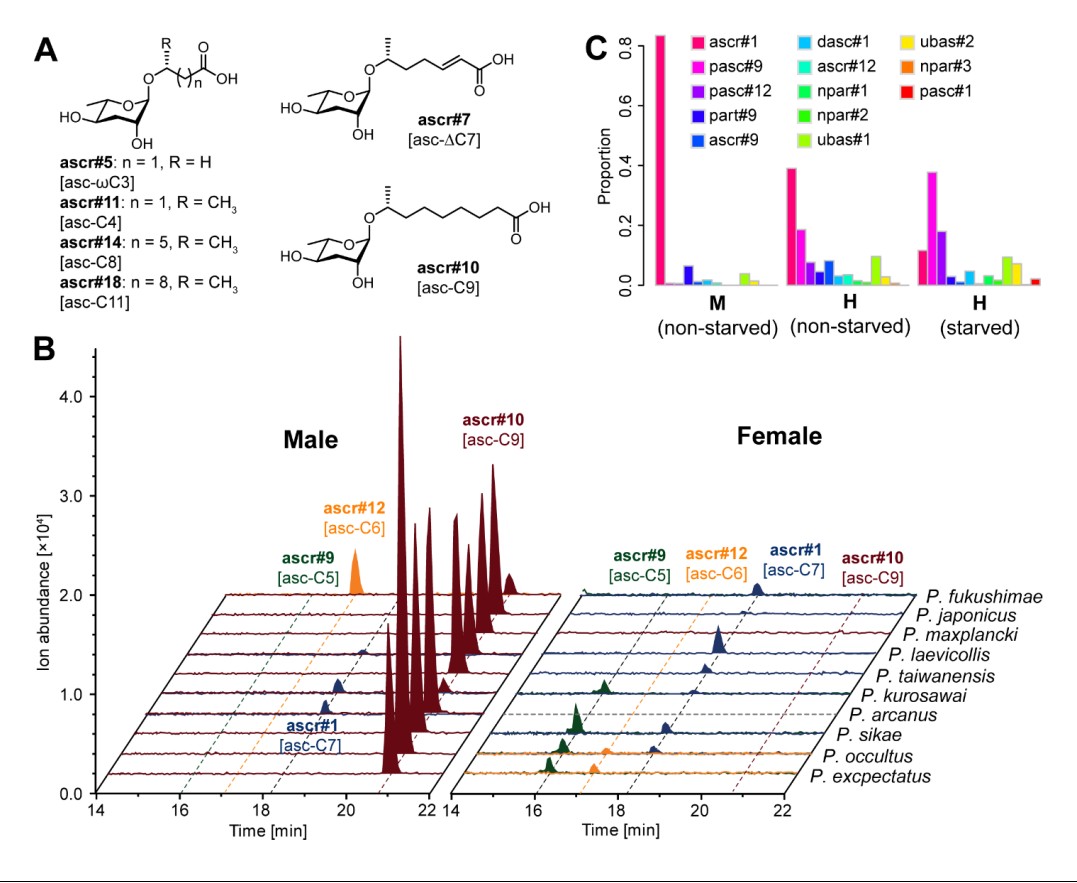

**Figure 3.** Simple ascarosides detected from *Pristionchus* nematodes. (**A**) Chemical structures of newly detected simple ascarosides. (**B**) Males of 10 gonochoristic species release large amounts of ascr#10 [asc-C9] in a sex-specific manner. Dashed gray line indicates that nothing was detected in the culture, probably because of contamination of the *P. arcanus* females. (**C**) Males and hermaphrodites of *P. pacificus* released ascr#1 [asc-C7] and pasc#9, respectively, but no ascr#10 [asc-C9]. M, male; H, hermaphrodite.

The online version of this article includes the following figure supplement(s) for figure 3:

**Figure supplement 1.** Simple ascarosides detected from various *Pristionchus* nematodes.

**Figure supplement 2.** ascr#18 [asc-C11] detected from *P. atlanticus*.

**Figure supplement 3.** ascr#7 [asc-ΔC7] was specifically detected from the nematodes of the *entomophagus* clade.

**Figure supplement 4.** Chemical composition of different sex-derived exo-metabolomes.

## *Pristionchus* nematodes produce diverse dimeric ascarosides

*P. pacificus* produces a dimeric ascaroside (dasc#1 [4'-(asc-C7)-asc-C7]) from two ascr#1 units that can pheromonally trigger development of the Eu mouth-form (*Figure 1B*; *Bose, 2012*). In theory, additional DASC compounds could be produced by the use of alternative simple ascarosides and/or alternative attachment positions. We detected dasc#1 from most *Pristionchus* species (24/32) alongside a number of additional DASC compounds that were formed from different simple ascarosides attached at either the 2'- or 4'-position of the ascarylose moiety (*Figures 2* and *4A– B*; *Supplementary file 3—Table 2*). These new DASCs were distributed across 27 of the 32 analyzed species, including dasc#1-producing *P. pacificus* and eight of the nine species for which dasc#1 was not detected. All told, DASC compounds were identified in 31 of 32 species, suggesting that this NDMM class originated prior to the diversification of the *Pristionchus* genus (*Supplementary file 4—Figure 2*). Most species produced multiple DASCs, among which dimers containing ascr#9 [asc-C5] and/or ascr#1 [asc-C7] were the most abundant (*Figure 4—figure supplement 1D*). This pattern is presumably due, at least in part, to the fact that these two precursors are produced at high levels in most species (*Figure 2* and *Figure 2—source data 1*).

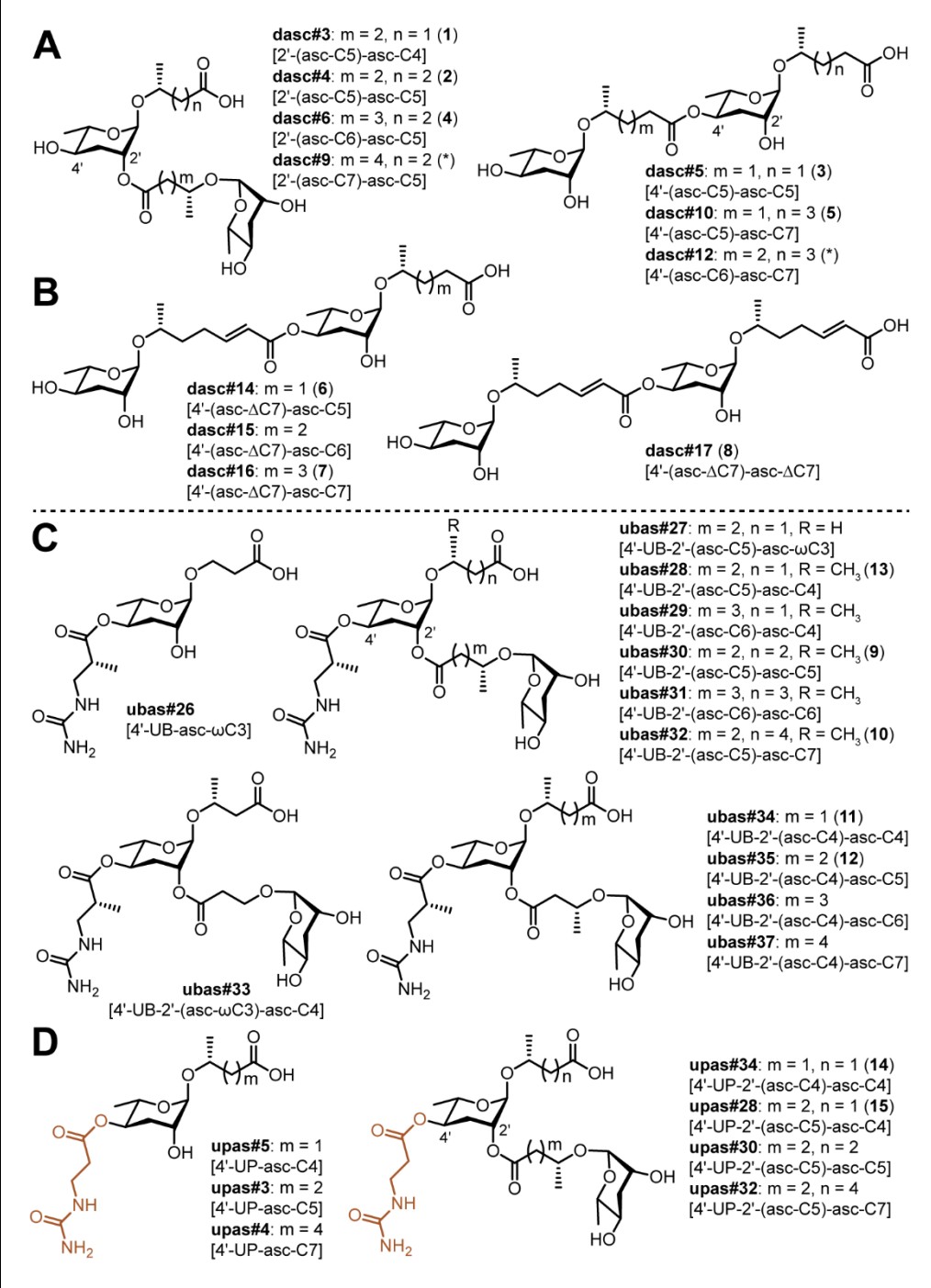

**Figure 4.** Novel DASC, UBAS, and UPAS chemicals detected from *Pristionchus* species. (**A**) Structures of new DASC chemicals identified from *Pristionchus* nematodes with linkage at the 2'- or 4'-position. Note that the asterisk-marked dasc#9 [2'-(asc-C7)-asc-C5] and dasc#12 [4'-(asc-C6)-asc-C7] were tentatively assigned as being linked at the 2'- or 4'-position, respectively. (**B**) Nematodes from the *entomophagus* clade specifically produced four novel DASC chemicals harboring one or two units of asc#7 [asc-ΔC7]. (**C**) Structures of novel UBAS chemicals identified from *Pristionchus* nematodes. (**D**) Newly identified UPAS chemicals carrying a ureidopropionic acid group at the 4'-position of ascarylose. For further details on NDMM nomenclature, see 'Materials and methods'.

The online version of this article includes the following figure supplement(s) for figure 4:

**Figure supplement 1.** Structure elucidation of DASC chemicals.

*Figure 4 continued on next page*

*Figure 4 continued*

**Figure supplement 2.** NDMMs detected from the *P. pseudoaerivorus* exo-metabolome.
**Figure supplement 3.** DASC chemicals detected from nematodes in the *maupasi* clade.
**Figure supplement 4.** DASC chemicals detected from nematodes in the *entomophagus* clade.
**Figure supplement 5.** Structures of five unquantifiable DASC chemicals detected from *Pristionchus* species.

---

To elucidate the chemical structures of newly identified DASCs, a combination of MS and NMR techniques was applied. Specifically, we used LC-ESI-(+)-MS/MS to cleave the terminal side chain to support the assignments of individual ascaroside units (*Figure 4—figure supplement 1C*), and we used 2D NMR (*dqf*-COSY) data to clarify whether the linkage involved the 2'- or 4'-position. For example, LC-ESI-(+)-MS/MS fragmentation of dasc#3 [2'-(asc-C5)-asc-C4, **1**], detected in *P. degawai* and *P. fukushimae* (*Figures 2* and *4A*), afforded an ion signal of $C_{17}H_{28}NaO_8^+$ (*m/z* 383.1667, [M + Na]$^+$, Δ 2.4 ppm) with a loss of the C4 side chain (*Supplementary file 2a—Figure 1*), which helped to assign the first and the second ascarosides as ascr#9 [asc-C5] and ascr#11 [asc-C4], respectively (see *Figure 4—figure supplement 1A-B* for details on the nomenclature of DASC chemicals). Subsequent analysis of the *dqf*-COSY spectrum of dasc#3 (**1**) from *P. fukushimae* corroborated these assignments and further demonstrated that ascr#9 was connected to the 2'-position of ascr#11 (*Supplementary file 1b—Figure 1* and *Supplementary file 3—Table 7*). Interestingly, we found that two ascr#9 units can be linked at the 2'- or 4'-position, resulting in two isomers, dasc#4 [2'-(asc-C5)-asc-C5, **2**] (*Figures 2*, *4A* and *5A*; *Supplementary file 3—Table 8*) and dasc#5 [4'-(asc-C5)-asc-C5, **3**] (*Figures 2*, *4A* and *5B*; *Supplementary file 3—Table 9*), the latter of which displayed a high specificity to nematodes of the *maupasi* clade (*Figure 4—figure supplements 2* and *3*). These findings revealed a rich chemical diversity of previously unknown DASC chemicals in *Pristionchus* nematodes.

Further chemical analysis revealed several homologous series of DASC chemicals in *Pristionchus* nematodes. The first series includes dasc#4 [2'-(asc-C5)-asc-C5, **2**], dasc#6 [2'-(asc-C6)-asc-C5, **4**], and dasc#9 [2'-(asc-C7)-asc-C5], which incorporate different simple ascarosides at the 2'-position of ascr#9 [asc-C5] (*Figure 4A*; *Supplementary file 2a—Figures 7–8*; *Supplementary file 3—Table 10*). Another homologous series of DASC chemicals includes dasc#10 [4'-(asc-C5)-asc-C7, **5**], dasc#12 [4'-(asc-C6)-asc-C7], and dasc#1 [4'-(asc-C7)-asc-C7] (*Figure 4A*; *Supplementary file 2a—Figures 9–11*; *Supplementary file 3—Table 11*), which each have a simple ascaroside attached to the 4'-position of ascr#1 [asc-C7]. Finally, we found four new dimeric ascarosides containing at least one ascr#7 [asc-ΔC7] unit, namely dasc#14 [4'-(asc-ΔC7)-asc-C5, **6**], dasc#15 [4'-(asc-ΔC7)-asc-C6], dasc#16 [4'-(asc-ΔC7)-asc-C7, **7**], and dasc#17 [4'-(asc-ΔC7)-asc-ΔC7, **8**] (*Figures 2* and *4B*). These DASC compounds were exclusively detected in species from the *entomophagus* clade, the only group in which the ascr#7 precursor was found (*Figure 4—figure supplement 4*). Here, LC-ESI-(+)-MS/MS analysis generated a fragment ion signal of $C_{19}H_{30}NaO_8^+$ (*m/z* 409.1865, [M + Na]$^+$, Δ 2.6 ppm) upon cleavage of the terminal side chains, indicating that ascr#7 is the first ascaroside in dasc#14–16 (*Supplementary file 2a—Figures 17–19*). According to the MS/MS analysis, the fourth dimer, dasc#17 (**8**), probably contained two identical ascr#7 units (*Supplementary file 2a—Figure 20*). To corroborate these assignments, reverse-phase C18 solid phase extraction (SPE) and HPLC were employed to purify dasc#14 (**6**) and dasc#17 (**8**) from *P. entomophagus* and *P. uniformis*, respectively. NMR analysis demonstrated the presence of one unsubstituted ascarylose moiety, one 4'-substituted ascarylose moiety, a (ω−1)-oxygenated C5 side chain and an unsaturated (ω−1)-oxygenated C7 side chain in dasc#14 (**6**), suggesting that ascr#7 [asc-ΔC7] was linked to the 4'-position of ascr#9 [asc-C5] (*Figure 5C* and *Supplementary file 3—Table 12*). 2D NMR data for dasc#17 (**8**) confirmed that two identical ascr#7 units were connected at the 4'-position (*Figure 5C* and *Supplementary file 3—Table 13*). Furthermore, analysis of the *dqf*-COSY spectrum of an enriched mixture of dasc#16 (**7**) and dasc#17 (**8**) indicated that ascr#7 [asc-ΔC7] was also linked to the 4'-position of ascr#1 [asc-C7] in dasc#16 (*Supplementary file 1b—Figure 47*). Thus, these four DASCs all have a linkage at the 4'-position. These results show how the production of novel simple NDMMs (i. e., ascr#7, which is found only in the *entomophagus* clade) can underlie the production of related modular compounds, thereby demonstrating linked innovation at multiple structural levels.

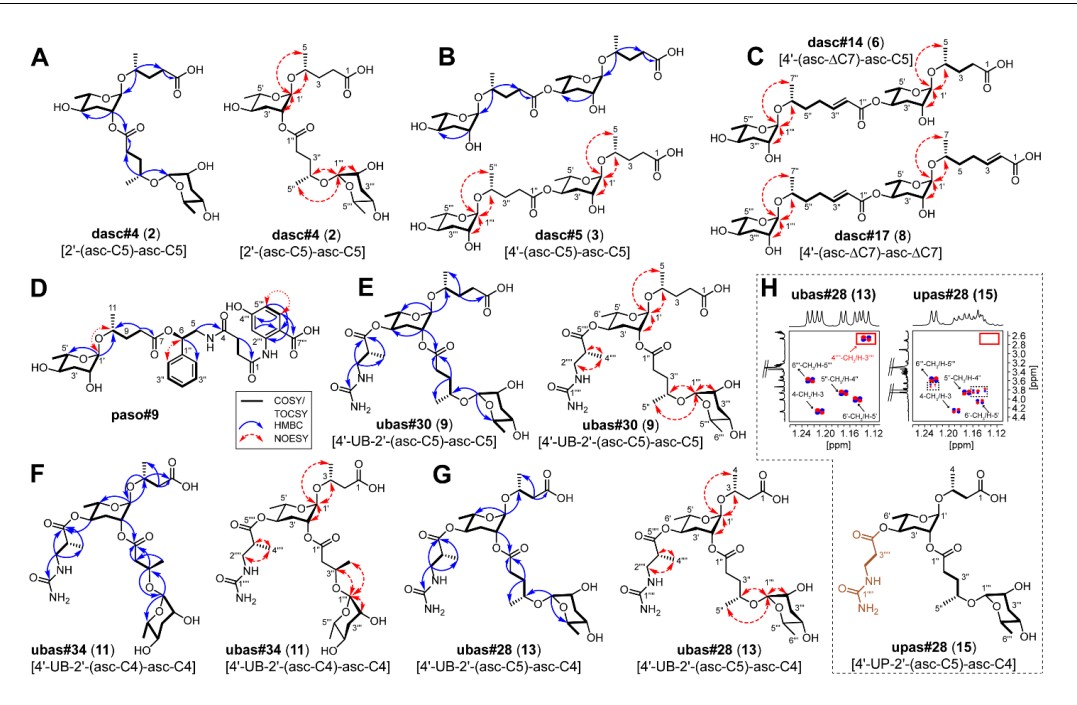

**Figure 5.** Elucidation of the structures of novel NDMMs. (**A**) dasc#4 [2'-(asc-C5)-asc-C5, **2**], (**B**) dasc#5 [4'-(asc-C5)-asc-C5, **3**], (**C**) dasc#14 [4'-(asc-ΔC7)-asc-C5, **6**], dasc#17 [4'-(asc-ΔC7)-asc- ΔC7, **8**], (**D**) paso#9, (**E**) ubas#30 [4'-UB-2'-(asc-C5)-asc-C5, **9**], (**F**) ubas#34 [4'-UB-2'-(asc-C4)-asc-C4, **11**], (**G**) ubas#28 [4'-UB-2'-(asc-C5)-asc-C4, **13**], and (**H**) upas#28 [4'-UP-2'-(asc-C5)-asc-C4, **15**]. Key COSY/TOCSY, NOESY, and/or HMBC correlations of each compound are summarized. Comparison of selected *dqf*-COSY spectra between ubas#28 (**13**) and upas#28 (**15**), highlighting correlations with methyl groups, indicates that upas#28 (**15**) contains a ureidopropionic acid moiety (**H**). The dashed boxes in panel (**H**) signal impurities, and the red-lined box highlights the absence of a methyl group in upas#28 (**15**). For details, see *Supplementary file 1d—Figure 38* and *Supplementary file 1e—Figure 2*.

The online version of this article includes the following figure supplement(s) for figure 5:

**Figure supplement 1.** Structural characterization of paso#9.
**Figure supplement 2.** MS/MS fragmentation pattern of UBAS chemicals.
**Figure supplement 3.** NDMMs detected from the *P. hoplostomus* exo-metabolome.
**Figure supplement 4.** UPAS chemicals detected from five *Pristionchus* species.
**Figure supplement 5.** MS/MS fragmentation pattern of UPAS chemicals.

## *P. pacificus* and its close relatives produce a diverse suite of structurally complex NDMMs

In addition to its simple and dimeric NDMMs, *P. pacificus* produces a variety of more structurally complex compounds—the NPAR, PASC and UBAS NDMMs (*Figure 1B*)—some of which are known to act as pheromonal regulators of dauer formation and mouth-form development (*Bose, 2012*; *Bose et al., 2014*; *Yim et al., 2015*). These three classes are defined by the presence of additional structural components that are attached to the sugar and the fatty acid side chain. The additional components derive from several different metabolic pathways, ranging from nucleotide to amino acid metabolism. Characterizing the phylogenetic history of these highly modular compounds is a key step in revealing the origins of NDMM complexity.

NPAR compounds, which are distinguished by the presence of nucleoside- and amino acid-containing sidechains and by the use of paratose as the sugar scaffold, were observed in the exo-metabolomes of *P. pacificus* and its six closest relatives, with npar#1 typically present at the highest relative levels (*Figure 2*). This distribution closely but imperfectly matches that of the NPAR precursor molecule, the simple paratoside part#9, with seven of the eight *pacificus*-clade species that produce part#9 also producing NPAR NDMMs. Ancestral reconstruction analyses point to a single origin for NPAR compounds that coincided with one of the two predicted origins for part#9,

suggesting that part#9 production is a necessary but not sufficient condition for NPAR production in *Pristionchus* nematodes (*Figure 6A* and *Supplementary file 4 - Figure 2*).

The fatty acid side chains of PASC NDMMs link to components that are derived from neurotransmitter metabolism and the citric acid cycle. PASC compounds were produced consistently by *P. pacificus* and its nine closest relatives, with pasc#9 being the most abundant compound in each case (*Figure 2* and *Supplementary file 1c—Figures 1–4*). Among the five PASC compounds detected in the present study were two pentamodular NDMMs, pasa#9 (*Yim et al., 2015*) and paso#9, which both include anthranilic acid as an additional side chain moiety. The production of paso#9 by *P. pacificus* was previously demonstrated by *Artyukhin et al., 2018* but its structure was incompletely characterized; our NMR analysis indicated the presence of a 4-hydroxyanthranilic acid moiety in paso#9 (*Figures 2* and *5D*; *Figure 5—figure supplement 1*; *Supplementary file 2b—Figure 1*; *Supplementary file 3—Table 14*).

UBAS is a diverse class of NDMMs defined by the presence of ureidoisobutyric acid at the 4'-position of ascarylose (*Bose, 2012*; *Artyukhin et al., 2018*; *Falcke et al., 2018*) and, in some cases (the dimeric UBASs), by the presence of an additional ascaroside unit that is attached as a side chain at the 2'-position. Our comparative screen uncovered several novel compounds and showed that UBAS is produced by most members of the *pacificus* clade, with a broader but patchier phylogenetic

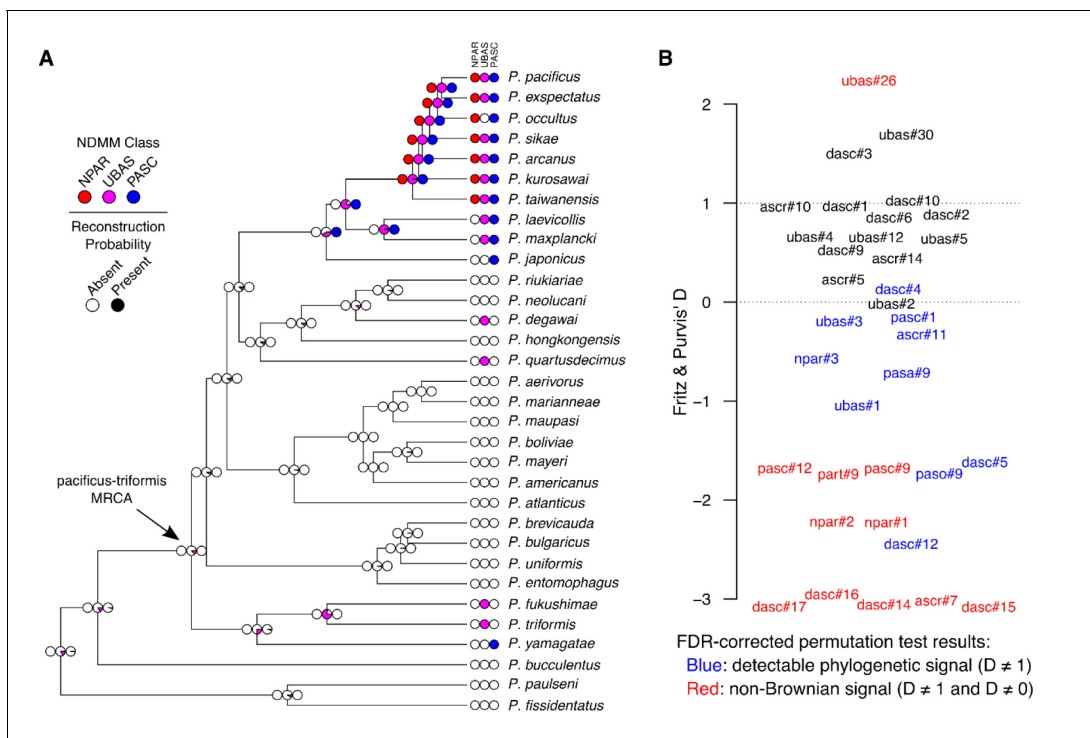

**Figure 6.** Comparative analysis of NDMMs in *Pristionchus*. (**A**) Convergent evolution of PASC and UBAS-type NDMMs in the *pacificus* and *triformis* clades. The evolution of complex NDMMs was reconstructed via Bayesian stochastic character mapping. Colored circles at the tips of the phylogeny denote the presence (dark color) or absence (white) of NPAR (red), UBAS (magenta), and PASC (blue) NDMMs; those at the internal nodes denote the probability of presence versus absence. Results indicate that NPAR compounds evolved only once (within the *pacificus* clade), whereas both PASC and UBAS compounds evolved convergently within the *pacificus* and *triformis* clades. The estimated probability of being present at the node representing the most recent common ancestor (MRCA) of the *pacificus* and *triformis* clades is less than 0.01 for both NPAR and PASC, and 0.12 for UBAS. Ancestral character mapping results for individual NDMMs are provided in *Table 1* and *Figure 6—figure supplement 1*. (**B**) Variation in phylogenetic signal strength among *Pristionchus* NDMMs. Binary-trait phylogenetic signal (Fritz and Purvis' D) was estimated for all non-constant, non-singleton NDMMs. D values of 0 and 1 correspond to expectations under Brownian evolution and non-phylogenetic randomness, respectively. Red labels denote NDMMs with D significantly different from both 0 and 1, whereas blue labels indicate those with D significantly different from 1 but not 0, on the basis of the results of permutation test-derived FDR corrected (10% threshold) *P* values. The remainder (black) represent cases in which we were unable to reject either baseline hypothesis. D-value estimates and permutation *P* values are provided in *Table 1*.

The online version of this article includes the following figure supplement(s) for figure 6:

**Figure supplement 1.** Comparative analysis of *Pristionchus* NDMM profiles.

**Table 1.** Mode of evolution of major *Pristionchus* NDMMs.

| NDMM | Class | Conservation Pattern | fitMk model | SIMMAP changes | D | P (Ho: D = 0) | P (Ho: D = 1) | Phylogenetic interpretation |
|---|---|---|---|---|---|---|---|---|
| ascr#1 | Simple NDMM | Constant | – | – | – | – | – | Strictly conserved |
| ascr#10 | Simple NDMM | Variable | ARD | 17.50 | 0.9645 | 0.2524 | 0.8989 | Variable |
| ascr#11 | Simple NDMM | Variable | ER | 15.71 | −0.3263 | 0.6697 | 0.0051 | Somewhat conserved (Brownian?) |
| ascr#12 | Simple NDMM | Constant | – | – | – | – | – | Strictly conserved |
| ascr#14 | Simple NDMM | Variable | ER | 26.25 | 0.4424 | 0.6116 | 0.2027 | Variable |
| ascr#5 | Simple NDMM | Variable | ER | 8.54 | 0.2276 | 0.8670 | 0.1933 | Variable |
| ascr#7 | Simple NDMM | Variable | ER | 1.10 | −3.0185 | 0.0854 | 0.0000 | Highly conserved |
| ascr#9 | Simple NDMM | Constant | – | – | – | – | – | Strictly conserved |
| part#9 | Simple NDMM | Variable | ER | 2.72 | −1.7518 | 0.0854 | 0.0000 | Highly conserved |
| dasc#1 | DASC | Variable | ER | 15.77 | 0.9714 | 0.2487 | 0.8989 | Variable |
| dasc#10 | DASC | Variable | ARD | 24.71 | 1.0261 | 0.1928 | 0.9866 | Variable |
| dasc#12 | DASC | Variable | ER | 1.64 | −2.4439 | 0.2335 | 0.0035 | Somewhat conserved (Brownian?) |
| dasc#14 | DASC | Variable | ER | 1.17 | −3.0560 | 0.0854 | 0.0000 | Highly conserved |
| dasc#15 | DASC | Variable | ER | 1.16 | −3.0805 | 0.0854 | 0.0000 | Highly conserved |
| dasc#16 | DASC | Variable | ER | 1.25 | −2.9528 | 0.0993 | 0.0000 | Highly conserved |
| dasc#17 | DASC | Variable | ER | 1.08 | −3.0772 | 0.0854 | 0.0000 | Highly conserved |
| dasc#2 | DASC | Variable | ER | 2.50 | 0.8850 | 0.6697 | 0.8989 | Variable |
| dasc#3 | DASC | Variable | ER | 2.66 | 1.5025 | 0.4077 | 0.7440 | Variable |
| dasc#4 | DASC | Variable | ER | 18.90 | 0.1325 | 0.8670 | 0.0644 | Somewhat conserved (Brownian?) |
| dasc#5 | DASC | Variable | ARD | 6.83 | −1.6165 | 0.2487 | 0.0032 | Somewhat conserved (Brownian?) |
| dasc#6 | DASC | Variable | ER | 37.88 | 0.8552 | 0.2487 | 0.7392 | Variable |
| dasc#9 | DASC | Variable | ARD | 17.90 | 0.5267 | 0.6116 | 0.3702 | Variable |
| npar#1 | NPAR | Variable | ER | 1.17 | −2.2388 | 0.0854 | 0.0000 | Highly conserved |
| npar#2 | NPAR | Variable | ER | 1.19 | −2.2318 | 0.0854 | 0.0000 | Highly conserved |
| npar#3 | NPAR | Variable | ARD | 8.11 | −0.5835 | 0.6463 | 0.0644 | Somewhat conserved (Brownian?) |
| pasa#9 | PASC | Variable | ARD | 7.16 | −0.7273 | 0.6116 | 0.0254 | Somewhat conserved (Brownian?) |
| pasc#1 | PASC | Variable | ARD | 7.89 | −0.1643 | 0.8601 | 0.0644 | Somewhat conserved (Brownian?) |
| pasc#12 | PASC | Variable | ER | 2.52 | −1.6914 | 0.0513 | 0.0000 | Highly conserved |
| pasc#9 | PASC | Variable | ER | 2.52 | −1.6928 | 0.0513 | 0.0000 | Highly conserved |
| paso#9 | PASC | Variable | ARD | 4.19 | −1.7483 | 0.1108 | 0.0000 | Somewhat conserved (Brownian?) |
| ubas#1 | UBAS | Variable | ER | 5.94 | −1.0425 | 0.2825 | 0.0006 | Somewhat conserved (Brownian?) |
| ubas#12 | UBAS | Variable | ER | 5.99 | 0.6570 | 0.6116 | 0.6005 | Variable |
| ubas#2 | UBAS | Variable | ARD | 11.46 | −0.0160 | 0.9172 | 0.1368 | Variable |
| ubas#26 | UBAS | Variable | ARD | 6.37 | 2.2410 | 0.0513 | 0.0993 | Over-dispersed |
| ubas#27 | UBAS | Singleton | – | – | – | – | – | Young |
| ubas#28 | UBAS | Singleton | – | – | – | – | – | Young |
| ubas#29 | UBAS | Singleton | – | – | – | – | – | Young |
| ubas#3 | UBAS | Variable | ARD | 11.67 | −0.1931 | 0.8262 | 0.0255 | Somewhat conserved (Brownian?) |
| ubas#30 | UBAS | Variable | ER | 2.66 | 1.6939 | 0.2825 | 0.6054 | Variable |

*Table 1 continued on next page*

*Table 1 continued*

| NDMM | Class | Conservation Pattern | fitMk model | SIMMAP changes | D | P (Ho: D = 0) | P (Ho: D = 1) | Phylogenetic interpretation |
|------|-------|----------------------|-------------|----------------|---|---------------|---------------|------------------------------|
| ubas#32 | UBAS | Singleton | – | – | – | – | – | Young |
| ubas#33 | UBAS | Singleton | – | – | – | – | – | Young |
| ubas#34 | UBAS | Singleton | – | – | – | – | – | Young |
| ubas#35 | UBAS | Singleton | – | – | – | – | – | Young |
| ubas#36 | UBAS | Singleton | – | – | – | – | – | Young |
| ubas#4 | UBAS | Variable | ER | 6.01 | 0.6606 | 0.6116 | 0.6005 | Variable |
| ubas#5 | UBAS | Variable | ER | 9.22 | 0.6413 | 0.6007 | 0.5546 | Variable |

distribution than was seen for the PASC and NPAR classes (*Figure 2*). LC-ESI-(+)-MS/MS analysis of known UBAS chemicals yielded a characteristic fragment ion signal of $C_{11}H_{19}N_2O_5^+$ (*m/z* 259.1334, [M + H]$^+$) containing a ureidoisobutyric acid moiety and an ascarylose unit (*Figure 5—figure supplement 2A–C*). This prompted us to perform a targeted MS/MS screen (*Figure 5—figure supplement 2D*; *Dong et al., 2016*) that enabled the identification of a series of novel UBAS chemicals from *pacificus*-clade species and, unexpectedly, from *triformis*-clade species, including ubas#26 (which lacks a 2′-linked sidechain) and the dimeric UBASs ubas#27–#37 (defined by the presence of various 2′-linked ascaroside side chains) (*Figures 2* and *4C*; *Supplementary file 3—Table 3*). Representative novel UBAS chemicals, including ubas#30 [4′-UB-2′-(asc-C5)-asc-C5, **9**] (*Figure 5E* and *Supplementary file 3—Table 15*), ubas#34 [4′-UB-2′-(asc-C4)-asc-C4, **11**] (*Figure 5F* and *Supplementary file 3—Table 16*), and ubas#28 [4′-UB-2′-(asc-C5)-asc-C4, **13**] (*Figure 5G* and *Supplementary file 3—Table 18*), were identified from *P. maxplancki*, *P. quartusdecimus*, and *P. fukushimae*. Biosynthesis of these novel dimeric UBAS chemicals was correlated to high levels of production of simple UBAS precursors such as ubas#3 [4′-UB-asc-C5] and ubas#5 [4′-UB-asc-C4] (*Figure 2* and *Supplementary file 1d—Figures 30–35*). These new discoveries, combined with our findings on DASC compounds (described above), highlight dimerization as an important mechanism for expanding the structural diversity of NDMMs.

## Convergent evolution of UBAS and PASC chemicals in *Pristionchus* nematodes

NPAR, PASC, and UBAS chemicals were not detected in the *maupasi* and *entomophagus* clades, nor were they found in any of the outgroup *Pristionchus* species (*Figure 2*). However, UBAS and PASC compounds were unexpectedly also found in members of the *triformis* clade. With regard to PASC compounds, we found that *P. yamagatae* produced pasc#9, pasc#12, and pasc#1. With regard to UBAS compounds, we found that *P. triformis* and *P. hoplostomus* produced a variety of monomeric UBAS NDMMs and that *P. fukushimae* produced a mixture of monomeric and dimeric compounds, some of which were entirely novel (*Figures 2* and *4C*; *Figure 5—figure supplement 3*). These phylogenetic patterns could conceivably be explained by the production of PASC and UBAS compounds in the most recent common ancestor of the *pacificus* and *triformis* clades, followed by multiple loss events in other *Pristionchus* clades. However, ancestral reconstruction analyses of both individual NDMMs and overall NDMM classes strongly suggest that neither PASC nor UBAS compounds were produced by this ancestral species (*Figure 6A* and *Supplementary file 4—Figure 2*). Rather, PASC compounds were predicted to have evolved twice (once each within the *pacificus* and *triformis* clades), whereas UBAS was predicted to have evolved four times (once within the *triformis* clade and three times within the *pacificus* clade) (*Figure 6*). These analyses therefore argue for the convergent evolution of highly modular PASC and UBAS NDMMs in the distantly related *pacificus* and *triformis* clades.

To shed greater light on the ancestral state for *Pristionchus* nematodes, we characterized NDMM production in several non-*Pristionchus* diplogastrid species (*Figure 7* and *Figure 7—figure supplement 1*). Specifically, we investigated a single species from each of *Parapristionchus* (the sister group to *Pristionchus*), *Micoletzkya*, *Fuchsnema*, *Diplogasteroides*, *Acrostichus*, and *Allodiplogaster*, thereby covering the deepest phylogenetic splits within the Diplogastridae family (*Susoy et al.,*

*2015*). Doing so uncovered a variety of simple and DASC NDMMs, supporting the notion that these compounds originated prior to the establishment and diversification of the *Pristionchus* genus. However, NPAR, PASC, and UBAS chemicals were not detected in any of the assayed species, even at trace levels, indicating that these highly modular compounds are *Pristionchus*-specific evolutionary innovations. Negative findings are of course sensitive to detection limits, but given the overall power of our analytical platform and the fact that we replicated this search over six outgroup species, we take the apparent lack of NPAR, PASC, and UBAS NDMMs in non-*Pristionchus* diplogastrids to indicate that these highly modular compounds are *Pristionchus*-specific evolutionary innovations.

## Comparative analysis of NDMMs in *Pristionchus*

We quantified phylogenetic signal in NDMM presence/absence for the 35 non-constant and non-singleton NDMMs (*Supplementary file 4—Figures 1 and 2*) reported in *Figure 2* using the binary-trait D statistic of *Fritz and Purvis, 2010*. D values of 1 are expected under random distributions, whereas D values of 0 are expected if the trait evolves according to a Brownian process. D was significantly different from one for 21 of 35 compounds, indicating the presence of phylogenetic signal for most *Pristionchus* NDMMs (*Figure 6B*). Of these, D was significantly less than zero for 10, identifying cases where the action of stabilizing selection has been particularly strong (i.e., stronger than expected under Brownian evolution).

We then tested for phylogenetic signal in overall NDMM profiles by examining whether multivariate phenotypic distances between NDMM profiles correlated with pairwise phylogenetic distances. Differences among NDMM profiles were quantified by calculating Bray-Curtis dissimilarities using the compositional data shown in *Figure 2* (but standardizing to 100% across all classes rather than within each class), whereas phylogenetic distances were extracted from an ultrametric rescaling of the *Pristionchus* phylogeny shown in *Figure 1—figure supplement 1*. Ordination of NDMM profile dissimilarities via non-metric multidimensional scaling was indicative of phylogenetic signal, returning more-or-less distinct clusters for the *entomophagus* clade, the *maupasi* clade, and the *Pristionchus* outgroups (*Figure 6—figure supplement 1A*). The distantly related *pacificus* and *triformis* clades, by contrast, overlapped with one another, suggesting convergence. Pairwise phenotypic dissimilarities were significantly positively related to pairwise phylogenetic distances (Mantel test: Spearman's rho = 0.308; p=0.002); the relationship was linear at lower evolutionary scales but plateaued when pairwise distances involved the distantly related outgroup species (*Figure 6—figure supplement 1B*). Notably, the intercept term for the linear fit was significantly greater than zero (95% CI = 0.354–0.433 when including outgroup comparisons, and 0.224–0.311 when ignoring outgroup comparisons; assuming independence among data points in both cases). A positive intercept implies that considerable phenotypic divergence is present at the point of speciation. Note that NDMM production rates are known to be sensitive to environmental factors (*Figure 3C*; *Kaplan et al., 2011*). We prepared nematode exo-metabolomes under identical laboratory conditions for all species, but because population growth rates varied among species, imposing standard culture conditions may have counterintuitively generated environmental variation in NDMM production rates. An exploration of the effects of culture conditions on NDMM diversity and evolution are beyond the scope of the present study but will be prioritized in future work.

## Trace amounts of pheromonal small molecules include a new family of UPAS chemicals from *Pristionchus* nematodes

Biochemical screens for pheromonal small molecules possess an inherent limitation, namely the detection limit imposed by the techniques used for biological sample collection and subsequent chemical analysis. We restricted our comparative analyses to consider only compounds detected at quantifiable levels (ion intensity of at least $1.0 \times 10^3$). However, we also detected the presence of several novel NDMMs in *Pristionchus* exo-metabolomes at unquantifiably low (trace) amounts, including one simple ascroside, five DASC NDMMs, two UBAS NDMMs, and a novel NDMM class that we refer to as UPAS (*Supplementary file 3—Table 5*). Reassuringly, the distribution of trace NDMMs does not appear to affect the general patterns identified when considering abundant NDMMs, most importantly the phylogenetically split distributions observed for UBAS and PASC compounds. The functional significance of NDMMs that are produced at such low levels is unclear and they may simply represent precursors or byproducts that are released unintentionally alongside

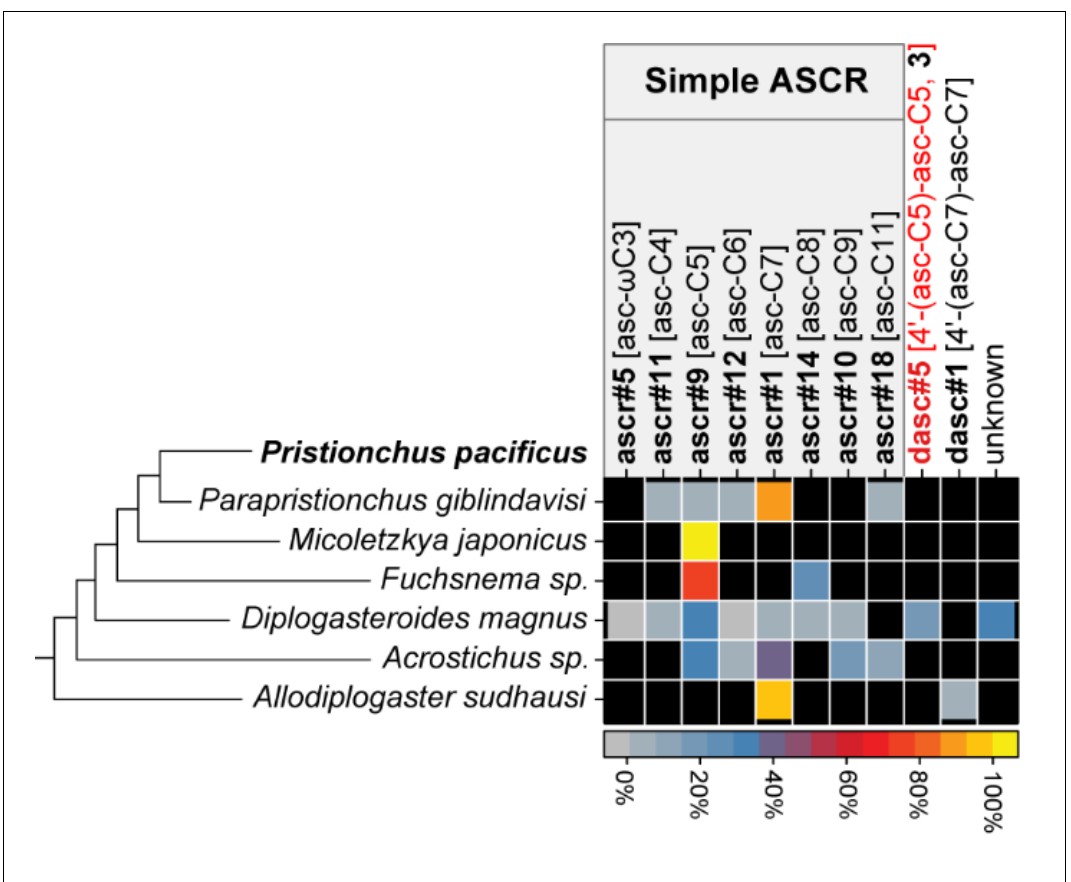

**Figure 7.** Summary of NDMMs in six (non-*Pristionchus*) Diplogastridae species that represent distant relatives of *P. pacificus*. Three biological replicates for each species were grown in 100 ml S-medium for LC/MS analysis (see 'Materials and methods'). Mean peak area for each compound from each species was divided by total peak area, thereby representing the abundance of each compound as a percentage (*Figure 7—source data 1*). Black color represents the apparent absence of the chemical. Black and red compound labels are known and novel NDMMs, respectively. 'Unknown' represents uncharacterized ascaroside.

The online version of this article includes the following source data and figure supplement(s) for figure 7:

**Source data 1.** Quantification of NDMMs in six (non-*Pristionchus*) Diplogastridae species.
**Figure supplement 1.** NDMMs detected from the *Diplogasteroides magnus* exo-metabolome.

more abundant NDMMs. Regardless, their identification provides yet more evidence for NDMM diversity in *Pristionchus*, and their structural characterization may prove useful for future studies focused on dissecting NDMM biosynthesis pathways. Structural details on the novel simple UBAS and DASC compounds can be found in *Figure 3—figure supplement 2*, *Figure 4C* and *Figure 4— figure supplement 5*. Here, we highlight the newly discovered UPAS class.

In several species, we found trace amounts of novel ascarosides that share very similar molecular formulas with UBAS chemicals (*Figure 5—figure supplement 4*). When performing LC-ESI-(-)-MS/ MS experiments, we observed a common fragment ion signal of $C_4H_7N_2O_3^-$ (*m/z* 131.0468, [M – H]$^-$) (*Figure 5—figure supplement 5*), suggesting that these compounds carry a ureidopropionic acid moiety at the 4'-position of ascarylose instead of ureidoisobutyric acid. Consistent with existing nomenclature rules, we named these as UPAS chemicals (*Figure 4D* and *Supplementary file 3— Table 4*; nomenclature of UPAS chemicals, see *Figure 5—figure supplement 5C* and 'Materials and methods').

To characterize these structures, two representative UPAS chemicals, upas#34 [4'-UP-2'-(asc-C4)- asc-C4, **14**] and upas#28 [4'-UP-2'-(asc-C5)-asc-C4, **15**] were enriched from *P. quartusdecimus* and *P. fukushimae*, respectively. Analysis of the *dqf*-COSY spectrum of upas#34 (**14**) confirmed a 2'- and 4'- substituted ascarylose unit, an unsubstituted ascarylose unit, a ureidopropionic acid moiety

(*Nair et al., 2010*; *Vogels and Van der Drift, 1976*), and two (ω−1)-oxygenated C4 side chains (*Figure 4D* and *Supplementary file 3—Table 19*). The presence of a ureidopropionic acid group ($C_4H_7N_2O_3^-$, *m/z* 131.0468, [M − H]$^-$) in upas#34 (**14**) was further confirmed by the comparative analysis of MS/MS fragmentation patterns between ubas#34 [4'-UB-2'-(asc-C4)-asc-C4, **11**] and upas#34 (**14**) (*Supplementary file 2c—Figure 9* and *Supplementary file 2d—Figure 4*). Similarly, MS/MS analysis of ubas#28 [4'-UB-2'-(asc-C5)-asc-C4, **13**] and upas#28 (**15**) suggested that upas#28 (**15**) also carries a ureidopropionic acid group, and both of these compounds possibly harbor an identical heterodimer consisting of ascr#11 [asc-C4] and ascr#9 [asc-C5] (*Supplementary file 2c—Figure 14* and *Supplementary file 2d—Figure 5*). Analysis of the *dqf*-COSY data of upas#28 (**15**) confirmed the presence of a 2'- and 4'-substituted ascarylose unit, an unsubstituted ascarylose unit, a ureido-propionic acid moiety, one (ω−1)-oxygenated C4 side chain, and one (ω−1)-oxygenated C5 side chain (*Figure 5H* and *Supplementary file 3—Table 20*). Further comparison of *dqf*-COSY spectra between ubas#28 (**13**) and upas#28 (**15**) clearly showed that the methyl group at C-3'''' is absent in upas#28 (**15**) (*Figure 5H*). However, chemical synthesis would be required to determine their exact structures. In summary, UPAS chemicals were, like UBAS NDMMs, detected within the phylogenetically separated *pacificus* and *triformis* clades, suggesting that they may be produced through a common biosynthetic pathway.

## Discussion

This study was designed to bring the comparative method (*Harvey and Pagel, 1991*) to the analysis of small molecule chemistry in *Pristionchus* nematodes. Previous studies indicated that the NDMM repertoire released by the model species *P. pacificus* differed substantially from that of *C. elegans*, but little was known about the underlying evolutionary dynamics. Given that *C. elegans* and *P. pacificus* belong to different nematode families and have been separated for roughly 100 million years, and the fact that two-species comparisons are inherently limiting, it was not possible to address the question of how NDMM biosynthesis evolves. However, findings from different research areas provided some initial insights into the patterns and processes of NDMM divergence. First, comparing different strains of *P. pacificus* revealed the existence of natural variation in NDMM production, thereby demonstrating the presence of the raw material needed for divergence among populations and, ultimately, species (*Bose et al., 2014*; *Falcke et al., 2018*). Second, comparative genomic studies comparing eight *Pristionchus* species revealed rapid gene turnover within gene families that are potentially involved in the biosynthesis of NDMMs (*Prabh et al., 2018*). Finally, many of the evolutionary and ecological processes that are at the focus of current research in *P. pacificus*, including competition and cheating with respect to the induction of dauer larvae, nematode predation, and self-recognition, revealed substantial cross-species interactions, suggesting the divergent evolution of signaling systems (*Bose et al., 2014*; *Mayer et al., 2015*; *Wilecki et al., 2015*; *Lightfoot et al., 2019*). These co-evolutionary and ecological processes might drive divergent evolution of NDMM structure, function, and biosynthesis. Building on these studies and determining the processes that shape the evolution of pheromonal systems requires an appreciation of the diversity of signaling molecules used among species. With this in mind, we set out to characterize patterns of NDMM production across the *Pristionchus* genus, thereby filling an important gap in our understanding of small molecule signaling in nematodes.

Consistent with expectations, our comparative analysis of 32 culturable *Pristionchus* species revealed considerable NDMM diversity. This diversity includes complex NDMMs that were previously only described from *P. pacificus*, as well as a wide variety of simple ascarosides that are found across the nematode phylum. Several simple ascarosides that were described before from *C. elegans*, yet were absent in *P. pacificus*, have now been found in other *Pristionchus* species. On top of this, we uncovered 16 novel DASC chemicals, 12 novel UBAS chemicals, and a new family of UPAS chemicals, thereby greatly expanding the catalog of complex NDMMs. These discoveries suggest the potential for communication via highly specific signaling channels in *Pristionchus* nematodes and, more broadly, indicate that biochemical characterization of additional nematode species will continue to uncover new dimensions of NDMM structural diversity.

The phylogenetic distribution of the major NDMMs within the *Pristionchus* genus, as summarized in *Figure 2*, allows several major conclusions. First, the core set of *Pristionchus* NDMMs consists of simple and DASC compounds: simple NDMMs are produced by all studied species, whereas DASCs

are produced by most, including multiple non-*Pristionchus* outgroup species, with multiple DASC compounds reconstructed at deep nodes within the *Pristionchus* phylogeny. Second, *Pristionchus* nematodes have expanded upon this core set of NDMMs in both lineage- and compound-specific ways. The highly modular NPAR, PASC, and UBAS NDMMs include various metabolic building blocks as accessory components that are attached to either the sugar or fatty acid moiety. These NDMMs all appear to be evolutionary innovations of the *Pristionchus* genus, being absent from outgroup *Pristionchus* species, other diplogastrids, and rhabditid nematodes such as *C. elegans*. On top of this, our results suggest that UBAS and PASC NDMMs are evolutionary novelties that, counterintuitively, evolved repeatedly, with both classes being found across several *pacificus*-clade species and some *triformis*-clade species. This finding prompts several questions about the functional role and biochemical synthesis of UBAS and PASC compounds. In terms of function, it will be interesting to see whether similar structures imply similar pheromonal roles: selection may have led to convergent structural evolution if these compounds target a conserved receptor or if the metabolic accessories that they incorporate provide information about conserved aspects of *Pristionchus* physiology. In terms of biochemistry, convergent production might indicate de novo evolution of biosynthetic pathways that are capable of combining UBAS and PASC precursor molecules in the two *Pristionchus* lineages. Alternatively, this could reflect the convergent repurposing of functionally similar enzymes. The latter idea would seem to be most parsimonious, although high levels of de novo gene evolution have been observed in *Pristionchus* nematodes (*Prabh et al., 2018*), suggesting that the former idea should not be ruled out. A related idea is that a single set of enzymes for producing UBAS and PASC NDMMs may have been present ancestrally but silenced and only reawakened in the *pacificus* and *triformis* lineages. This idea has been invoked to explain large-scale variation in the presence or absence of classes of signaling cuticular hydrocarbons in the Hymenoptera (*Kather and Martin, 2015*).

*Pristionchus* NDMMs evolved in distinct ways. This is clear from a visual inspection of their phylogenetic distribution and from formal estimation of the strength of phylogenetic signal via Fritz and Purvis' D statistic: some compounds are highly variable, with more-or-less random distributions (D ≈ 1), whereas others are more phylogenetically restricted (D < 1). Notably, the set of NDMMs that have particularly low D values (D < 0)—that is, compounds with highly constrained distributions—is comprised of three groups of structurally related compounds: (1) the unsaturated simple NDMM ascr#7 and the four ascr#7-containing DASC compounds (all found exclusively within the *entomophagus* clade); (2) the simple paratoside part#9 and its modular derivatives npar#1 and npar#2 (all found only within the *pacificus* clade); and (3) the PASC compounds pasc#9 and pasc#12 (both found within the *pacificus* clade and, separately, within *P. yamagatae* of the *triformis* clade). This finding suggests that phylogenetic signal for *Pristionchus* NDMM presence or absence is largely the result of clade-specific enzymatic innovation. Determining which enzymes are involved will be a challenge because of the fact that most of these compounds are restricted to *Pristionchus* nematodes, so that it is not straightforward to transfer insights from previous studies of NDMM production in *C. elegans*. However, the simple NDMM ascr#7 is produced by *C. elegans*, and in this case we speculate that similar molecular mechanisms are involved in the biosynthesis of its unsaturated C7 side chain, namely *acox* activity (*Butcher et al., 2009*; *von Reuss et al., 2012*). One option forward might be to combine insights gleaned from *P. pacificus*-centered genetic screens with phylostratigraphic data obtained from comparative genomic surveys (*Falcke et al., 2018*; *Rödelsperger et al., 2018*).

The dynamic evolutionary patterns seen for the different NDMM classes result in considerable variability among species: aside from the outgroup species *P. paulseni* and *P. bucculentus*, no two *Pristionchus* species produce fully compatible sets of compounds. Moreover, quantitative NDMM levels can be quite different even for species that produce qualitatively similar sets of compounds. This combination of qualitative (presence/absence) and quantitative (relative abundance) variation lead to high levels of species-specificity. Our results suggest that the evolution of highly specific NDMM profiles across the genus involved a mixture of gradual change and rapid divergence. On the one hand, the positive relationship between NDMM profile dissimilarity and phylogenetic distance appears to be largely linear (especially when discounting the divergent outgroup species), suggesting gradualism—as species became more evolutionarily distant, their NDMM profiles became less and less alike. On the other hand, the positive intercept term suggests that even young species pairs with negligible evolutionary distance would have distinct NDMM profiles. This sort of pattern

has been taken as evidence for a link between pheromonal divergence and speciation in ants (*van Wilgenburg et al., 2011*) and bark beetles (*Symonds and Gitau-Clarke, 2016*), the idea being that signaling systems involved in mate choice would be under strong selection to diverge so as to facilitate accurate reproductive decisions. Whether NDMMs contribute to mate signaling in *Pristionchus* has not yet been examined, but it seems plausible as NDMMs have been linked to sex-specific signaling in *Caenorhabditis* nematodes (*Leighton and Sternberg, 2016*). Together, these patterns suggest that future studies should place greater emphasis on the ways that NDMMs affect sex-specific signaling and reproductive biology, in particular on how this varies among closely related species. Research efforts in *Pristionchus* have primarily focused on hermaphroditic *P. pacificus*, but future exploration of sex-specific NDMMs would be best served through a focus on gonochoristic species in which mate signaling is presumably much more tightly linked to fitness. Indeed, recent studies on longevity revealed fundamental differences between hermaphroditic and gonochoristic *Pristionchus* species (*Weadick and Sommer, 2016a*; *Weadick and Sommer, 2016b*; *Weadick and Sommer, 2017*). Finally, it seems likely that the rapid evolution of pheromonal signaling during speciation would be facilitated by there being ample standing genetic variation in NDMM production rates, as has been previously documented in *P. pacificus* (*Bose et al., 2014*; *Falcke et al., 2018*). However, comparable studies of intraspecific NDMM diversity in gonochorists have not yet been attempted.

The identification and structural elucidation of many novel NDMMs, as provided here, stands to motivate multiple lines of inquiry. First, the assignment of biological functions to individual NDMMs requires their chemical synthesis, followed by appropriate bioassays of said compounds (both individually and in biologically relevant combinations). When the diversity of NDMMs structures was first discovered in *P. pacificus*, subsequent synthesis of selected NDMMs revealed functions for some but not all of the individual compounds, such as dasc#1 (mouth form development) and npar#1 (dauer development) (*Bose, 2012*). Second, the identification of the biosynthetic pathways that are involved in the generation of structurally diverse NDMMs will represent the major challenge for future research. In general, the elucidation of biosynthetic pathways is slow, but in principle possible. Previous work in *P. pacificus* revealed that linking patterns of genomic and metabolomic variation seem among natural isolates can uncover genes that are involved in the biosynthesis of UBAS chemicals (*Falcke et al., 2018*). However, such studies require multiple strains and the existence of sufficient natural variation in NDMM synthesis, a pre-requisite that might not be fulfilled for all compounds. Third, promiscuity is to be expected for some of the enzymes involved in NDMM production (*Panda et al., 2017*; *Zhou et al., 2018*). Therefore, some of the NDMMs that we uncovered might represent by-products that have little or no biological function. Indeed, previous studies on two *Ppa-daf-22* genes involved in β-oxidation and the *Ppa-uar-1* gene involved in UBASs biosynthesis revealed the production of so-called 'shunt metabolites', which are mostly seen in NDMM biosynthesis-defective animals, a phenomenon also known in *C. elegans* (*von Reuss et al., 2012*; *Markov et al., 2016*; *Falcke et al., 2018*). Another possibility is that certain NDMMs represent ancestral relics that used to be functional and are simply produced because of the inability of evolution to immediately and completely erase the past. Given the rapid turnover seen across the phylogeny, these relics might be a fairly important contributor to extant diversity patterns.

*Pristionchus* nematodes clearly produce an impressive diversity of small molecules, including numerous highly modular compounds that have never before been discovered in nature. The structural nature of these compounds may prove to be specific to *Pristionchus*, but the evolutionary processes that generated this complexity are likely to be shared broadly. Most of these compounds presumably play some role in pheromonal communication. We suspect that competitive interactions in the decaying scarab beetle ecosystem, where *Pristionchus* worms compete for food and mating partners alongside many other chemically communicating species (including other ascaroside-producing nematodes), promote co-evolutionary pheromonal diversification, similar to that shown for insects and plants that also harbor high levels of secreted small-molecule diversity (*Symonds and Elgar, 2008*; *Engl et al., 2018*; *Lombe et al., 2019*). From this, we speculate that small molecules are, in general, rapidly evolving in nematodes, albeit with lineage-specific axes of structural diversification (*Dong et al., 2016*; *Dong et al., 2018*; *Dolke et al., 2019*; *Bergame et al., 2019*). Such patterns are most probably also connected to the highly dynamic patterns of genome evolution seen in nematodes, which involve the rapid turnover of genes that are linked to small molecule biosynthesis. *Pristionchus* nematodes provide an ideal system for exploring the interrelated mechanisms—co-

evolutionary ecology-driven processes on the one hand and the dynamics of genome and gene family evolution on the other—that drive the rapid and, in some cases, repeated evolution of small molecule architecture.

# Materials and methods

## Key resources table

| Reagent type (species) or resource | Designation | Source or reference | Identifiers | Additional information |
|---|---|---|---|---|
| Strain, strain background (*Caenorhabditis elegans*) | N2 | Caenorhabditis Genetics Center | | |
| Strain, strain background (*Pristionchus pacificus*) | RS2333 | RJ Sommer Lab | | |
| Strain, strain background (*P. pacificus*) | RS5410 (Ex[*eud-1*]) | RJ Sommer Lab | | |
| Strain, strain background (*P. exspectatus*) | RS5522 | RJ Sommer Lab | | |
| Strain, strain background (*P. occultus*) | RS5811 | RJ Sommer Lab | | |
| Strain, strain background (*P. sikae*) | RS5901 | RJ Sommer Lab | | |
| Strain, strain background (*P. arcanus*) | RS5527 | RJ Sommer Lab | | |
| Strain, strain background (*P. kurosawai*) | RS5527 | RJ Sommer Lab | | |
| Strain, strain background (*P. taiwanensis*) | RS5797 | RJ Sommer Lab | | |
| Strain, strain background (*P. laevicollis*) | RS5939 | RJ Sommer Lab | | |
| Strain, strain background (*P. maxplancki*) | RS5594 | RJ Sommer Lab | | |
| Strain, strain background (*P. japonicus*) | RS5238 | RJ Sommer Lab | | |
| Strain, strain background (*P. riukiariae*) | RS5937 | RJ Sommer Lab | | |
| Strain, strain background (*P. neolucani*) | RS5949 | RJ Sommer Lab | | |
| Strain, strain background (*P. degawai*) | RS5938 | RJ Sommer Lab | | |
| Strain, strain background (*P. hongkongensis*) | RS5957 | RJ Sommer Lab | | |
| Strain, strain background (*P. quartusdecimus*) | RS5230 | RJ Sommer Lab | | |
| Strain, strain background (*P. pseudoaerivorus*) | RS5139 | RJ Sommer Lab | | |
| Strain, strain background (*P. aerivorus*) | RS5106 | RJ Sommer Lab | | |
| Strain, strain background (*P. marianneae*) | RS5108 | RJ Sommer Lab | | |
| Strain, strain background (*P. maupasi*) | RS0143 | RJ Sommer Lab | | |
| Strain, strain background (*P. boliviae*) | RS5518 | RJ Sommer Lab | | |
| Strain, strain background (*P. mayeri*) | RS5460 | RJ Sommer Lab | | |
| Strain, strain background (*P. americanus*) | RS5140 | RJ Sommer Lab | | |
| Strain, strain background (*P. atlanticus*) | CZ3975 | RJ Sommer Lab | | |
| Strain, strain background (*P. brevicauda*) | RS5231 | RJ Sommer Lab | | |
| Strain, strain background (*P. bulgaricus*) | RS5283 | RJ Sommer Lab | | |
| Strain, strain background (*P. uniformis*) | RS0141 | RJ Sommer Lab | | |
| Strain, strain background (*P. entomophagus*) | RS0144 | RJ Sommer Lab | | |
| Strain, strain background (*P. fukushimae*) | RS5595 | RJ Sommer Lab | | |
| Strain, strain background (*P. hoplostomus*) | JU1090 | RJ Sommer Lab | | |
| Strain, strain background (*P. triformis*) | RS5233 | RJ Sommer Lab | | |
| Strain, strain background (*P. yamagatae*) | RS5964 | RJ Sommer Lab | | |
| Strain, strain background (*P. bucculentus*) | RS5596 | RJ Sommer Lab | | |
| Strain, strain background (*P. paulseni*) | RS5918 | RJ Sommer Lab | | |
| Strain, strain background (*P. fissidentatus*) | RS5133 | RJ Sommer Lab | | |
| Strain, strain background (*Micoletzkya japonicus*) | RS5524B | RJ Sommer Lab | | |

*Continued on next page*

*Continued*

| Reagent type (species) or resource | Designation | Source or reference | Identifiers | Additional information |
|---|---|---|---|---|
| Strain, strain background (*Parapritionchus giblindavisi*) | RS5555B | RJ Sommer Lab | | |
| Strain, strain background (*Diplogasteroides magnus*) | RS5740 | RJ Sommer Lab | | |
| Strain, strain background (*Diplogasteroides magnus*) | RS1987 | RJ Sommer Lab | | |
| Strain, strain background (*Allodiplogaster sudhausi*) | SB413 | RJ Sommer Lab | | |
| Strain, strain background (*Fuchsnema* sp.) | RS5592 | RJ Sommer Lab | | |
| Strain, strain background (*Acrostichus* sp.) | RS5722 | RJ Sommer Lab | | |

## Nomenclature of ascarosides

Newly identified ascarosides were named using four-letter codes according to the nomenclature system for 'Nematode Derived Modular Metabolites' (NDMMs) established for the SMID database (www.smid-db.org), which is maintained by Frank C Schroeder and Lukas Mueller in collaboration with Wormbase (www.wormbase.org). Given that one DASC (dasc#1) and 25 UBAS chemicals were previously reported from *P. pacificus* (*Bose, 2012*; *Artyukhin et al., 2018*; *Falcke et al., 2018*), the newly identified DASC chemicals were named dasc#2–17 and, similarly, the newly identified UBAS chemicals were named ubas#26–37. When compared to the UBAS chemicals containing an ureidoisobutyric acid group at the 4'-position, newly identified UPAS chemicals that harbor the same ascaroside scaffold but that have an alternative ureidopropionic acid group at the 4'-position were named using the same number. For example, two different NDMMs containing the same unit of ascr#11 but different moieties of ureidoisobutyric acid and ureidopropionic acid at the 4'-position of ascr#11, were named as ubas#5 [4'-UB-asc-C4] and upas#5 [4'-UP-asc-C4], respectively. To describe easily the MS/MS fragmentation patterns of DASC, UBAS, and UPAS chemicals, structure-based abbreviations were also used to name these newly identified ascarosides as follows: (head group-)asc-(ω)(Δ)C#(terminal group). For complex NDMMs containing two simple ascarosides units, the ascarosides were labeled as first or second on the basis of their relative spatial positions (see *Figure 4—figure supplement 1A-B*; *Figure 5—figure supplement 2C*; and *Figure 5—figure supplement C*), and the first ascaroside was placed in parentheses to differentiate it from the second ascaroside.

## Nematode strains and maintenance

Nematodes listed in *Figure 1—figure supplement 1* were used for this study. Recipes for nematode growth medium (NGM) plates and the general maintenance of worms have been described previously (*Ogawa et al., 2009*).

## Chemicals and reagents

Synthetic ascarosides and paratosides were used as standards for chemical analysis (*Bose, 2012*; *Yim et al., 2015*). Celite for column chromatography was purchased from Macherey-Nagel, Germany. Previously described ascarosides and paratosides from *P. pacificus* and *C. elegans* are listed in *Supplementary file 3—Table 1*; newly identified DASC, UBAS, and UPAS chemicals are listed in *Supplementary file 3—Table 2–4*.

## Chromatographic spectrometers

LC-HR(ESI$^{-/+}$)-MS and LC-HR(ESI$^{-/+}$)-MS/MS analysis was performed using a Dionex Ultimate 3000 HPLC instrument coupled to a Bruker Impact II ultrahigh resolution qTOF mass spectrometer equipped with an electrospray ionization, operating in negative or positive ion mode. A chromatographic reverse phase C18 column (Agilent Eclipse XDB-C18, 250 × 4.6 mm, 5 µm) was used to separate chemicals. A mixture of water ($H_2O$ + 0.1% formic acid) and acetonitrile (ACN + 0.1% formic acid) was used to elute the LC-MS system with a flow rate of 0.4 ml/min. Electrospray ionization (ESI) conditions for the Impact II ultrahigh resolution qTOF system were end plate offset 500 V, capillary

voltage 4500 V, dry gas flow of 12.0 l/min and dry temperature 200℃. A sodium formate solution (250 ml isopropanol, 1 ml formic acid, 5 ml 1 M NaOH in 500 ml water) was used as calibration solution. LC-MS or LC-MS/MS analysis were conducted using the single MS mode or 'Auto MS/MS' mode by scanning from $m/z$ 50 to 1300, respectively. Compass DataAnalysis software was used to calibrate, process and analyze MS or $MS^2$ data. For initial fractionation of crude extracts, a reverse phase C18 SPE column (45 ml/5000 mg, Macherey-Nagel, Germany) was used with a mixture of $H_2O$ and MeOH as solvent. Target SPE fractions were further separated by a Bio-Brad semi-preparative HPLC instrument equipped with a reverse phase C18 column (Ascentis C18, 250 × 10 mm, 5 µm), which was eluted using a mixture of water ($H_2O$ + 0.1% formic acid) and acetonitrile (ACN + 0.1% formic acid) with a flow rate of 2.0 ml/min.

## NMR spectroscopy

All 1D and 2D NMR spectra of isolated or enriched ascarosides were acquired using a Bruker Avance III 800 MHz spectrometer equipped with a triple gradient x, y, z-TXI probe-head. Only parts of the NMR spectra of paso#9 (listed in *Supplementary file 1c—Figures 5–7*) were acquired using a Bruker Avance II 400 MHz spectrometer. All samples were dissolved in $CD_3OD$ and all NMR spectra were acquired at 298℃K (24.85℃). Chemical shifts were referenced to $\delta(C\underline{H}D_2OD)$ = 3.31 ppm and $\delta(\underline{C}HD_2OD)$ = 49.00 ppm. *dqf*-COSY, TOCSY and NOESY spectra were acquired with a relaxation delay of 1.8 s. 4096 ($t_2$) x 142 ($t_1$) complex points were used for the *dqf*-COSY; 1024 ($t_2$) x 256 ($t_1$) for the TOCSY (spin lock mixing time of 70 ms); and 2048 ($t_2$) x 128 ($t_1$) for the NOESY (NOE mixing time of 300 ms). Carbon shifts were determined from HSQC and HMBC spectra. Both HSQC and HMBC spectra were recorded with 1024 ($t_2$) x 128 ($t_1$) complex points using a relaxation delay of 1.5 s. Bruker Topspin four software was used to process and analyze acquired NMR data.

## Chemical analysis of exo-metabolomes derived from 100 males and females of gonochoristic species

To analyze the chemical composition of sex-specific excretomes, several gonochoristic species that were known to produce ascr#10 abundantly were selected for chemical analysis. 100 males and females were separately cultured in 100 µl M9 buffer for 24 hr without bacterial food. Worms were filtered, and the resulting supernatant was directly injected into LC-MS (with an injection volume of 20 µl) for chemical analysis and quantification of NDMMs (see *Figure 3B* and *Figure 3—figure supplement 4*).

## Chemical analysis of exo-metabolomes derived from 1000 *P. pacificus* males and hermaphrodites

One thousand *P. pacificus* RS2333 males or hermaphrodites were collected just prior to maturation and washed several times with sterile water before being transferred into a 1.5 ml microcentrifuge tube. Each tube contained sterilized 990 µl of tap water and 10 µl of freeze-killed OP50 in S-medium (0.2 g/ml), for a final concentration of 1 worm/µl. Tubes were incubated for approximately 20 hr at 20℃ (with constant agitation), after which the supernatant was isolated and frozen at −20 ℃ in preparation for chemical analysis. Three replicates each were prepared for males and hermaphrodites. Small molecule profiling of the sex-specific supernatants was carried out by selective ion monitoring (SIM)-HPLC-MS analysis (*von Reuss et al., 2012*) of NDMMs from *P. pacificus*. Small molecule profiles were estimated by calculating the integral of the (SIM)-HPLC-MS peak for each compound, divided by the grand within-sample total. We also obtained small molecule profile data for RS2333 worms cultured under dauer-inducing conditions. Relative to the sex-specific cultures described above, dauer-cultures have higher population densities and lower food supplies, resulting in prolonged starvation conditions (see *Figure 3C*).

## Preparation of *Escherichia coli* pellets as a food source for maintaining nematodes in liquid culture

LB medium (4.0 l) inoculated with a single colony of *E. coli* OP50 was incubated in a shaker at 37℃ with 170 rpm for one night. *E. coli* OP50 was harvested next morning through centrifugation (SLA-3000 rotor, 4250 rpm, 10 min, 4℃). *E. coli* OP50 pellets were transferred into 50 ml falcon tubes for further centrifugation (F13−14 × 50 CY rotor, 6000 rpm, 10 min, 4℃). The resulting *E. coli* OP50

pellets were equally split into six 50.0 ml falcon tubes and stored in the cold room (8°C) for further usage. *E. coli* OP50 pellets were diluted using M9 buffer and added into the nematode liquid cultures as a food source.

## Establishment of small-scale liquid culture (100 ml) of nematodes for comparative analysis

Worms from five large (10 cm diameter) agar plates were washed with 6 ml M9 buffer into 250 ml flasks containing 100 ml S-medium and diluted *E. coli* OP50 pellet. Nematodes in the liquid culture were grown in a shaker at 22°C and 120 rpm for two weeks. During the first week, *E. coli* OP50 pellets were diluted using 9 ml M9 buffer and added into the flasks as the food source at two-day intervals (1/3, 1/3, 1/2, and 1 tube of diluted *E. coli* OP50 pellets were added into each flask, respectively) (*Dong et al., 2016*). Worms were harvested after two weeks by centrifugation (F13−14 × 50 CY rotor, 6000 rpm, 10 min, 4°C). The resulting supernatants were frozen at −80°C for one night and further dried by lyophilization into solid powder, which was extracted using 3 × 100 ml MeOH. Crude extracts were combined, filtered and dried (40°C) under reduced pressure to remove MeOH. Samples were finally dissolved in 2.0 ml MeOH, and 100 µl aliquots were used for LC-MS analysis. In total, six culturable Diplogastrid species and 32 culturable *Pristionchus* species were grown as 100 ml liquid cultures for comparative LC/MS analysis. Three replicates for each species were established.

## Untargeted screening for ascarosides in established small-scale liquid culture extracts

Following initial processing of the small-scale liquid culture samples, untargeted NDMM screening was conducted via LC-MS using a small aliquot of 100 µl, an injection volume of 5 µl, and a flow rate of 0.4 ml/min. A mixture of solvents containing water ($H_2O$ + 0.1% formic acid) and acetonitrile (ACN + 0.1% formic acid) was used to wash the LC/MS system. During the first 3 min, 3% ACN was applied to wash and equilibrate the LC/MS system, after which gradient washing was applied with the ACN content increased from 3% to 100% over the following 30 min. Isocratic washing with 100% ACN was used to wash the LC/MS system for 5 min, after which the ACN content of the solvent was decreased to 3% within the next 5 min. MS detection was performed using the Bruker Impact II ultrahigh resolution qTOF system, which was equipped with an electrospray ionization (ESI) source operating in negative or positive ion mode. ESI conditions for the Impact II ultrahigh resolution qTOF system were end plate offset 500 V, capillary voltage 4500 V, dry gas flow of 12.0 l/min and dry temperature 200°C. A sodium formate solution was used as calibration solution. After every third sample, a blank MeOH run was performed using the same method. For LC-MS analysis, samples were analyzed by LC-MS using single MS mode by scanning from *m/z* 50 to 1300. MS/MS analysis was conducted using nitrogen as collision gas, and 'Auto MS/MS' mode was applied by scanning from *m/z* 50 to 1300. All of the exo-metabolome extracts (derived from 100 ml liquid culture) were comparatively and systematically analyzed using LC-MS and LC-MS/MS approaches in both positive and negative ion modes. MS or $MS^2$ data were calibrated, processed, and analyzed using the Compass DataAnalysis software.

## Establishment of large-scale liquid culture for compound isolation

Twelve species (*P. pacificus*, *P. taiwanensis*, *P. laevicollis*, *P. maxplancki*, *P. quartusdecimus*, *P. mayeri*, *P. bulgaricus*, *P. uniformis*, *P. entomophagus*, *P. fukushimae*, *P. triformis* and *D. magnus*) were cultured in 2.0 l S-medium for chemical purification or enrichment (*Supplementary file 3— Table 6*). Worms from each of 10 big (10 cm diameter) agar plates were washed with 12 ml M9 buffer into 500 ml flasks containing 200 ml S-medium and diluted *E. coli* OP50 pellets, resulting in a total of 2.0 l (10 flasks × 200 ml) liquid culture, which was incubated in a shaker at 22°C and 120 rpm for two weeks. During the first week, *E. coli* OP50 pellets were diluted using 9 ml M9 buffer and added into the flasks as a food source at two-day intervals (1/3, 1/3, 1/2 and 1 tube of diluted *E. coli* OP50 pellets were added into each flask, respectively) (*Dong et al., 2016*). Worms were harvested after two weeks by centrifugation (F13−14 × 50 CY rotor, 6000 rpm, 10 min, 4°C). The resulting supernatants were frozen at −80°C for one night and further dried by lyophilization into solid

powder, which was extracted using MeOH. Crude extracts were combined, filtered and dried (at 40°C) under reduced pressure to remove MeOH.

## Initial fractionation of established large-scale liquid culture extracts

Large-scale liquid culture (2.0 l) extracts were initially fractionated using a RP-C18 SPE column to enrich the target compounds for further purification (*Supplementary file 3—Table 6*). First, the RP-C18 SPE column was flushed with 20 ml MeOH followed by 60 ml distilled $H_2O$ before sample application. Crude extracts mixed with an equal weight of Celite (about 2.0 g) were loaded onto the RP-C18 SPE column, which was washed with 20 ml distilled $H_2O$. Next, 20 ml of solvent ($H_2O$ and MeOH) was applied to elute compounds, with the MeOH content progressively increased from 10% to 100% in increments of 10%. Eluents were collected in 10 fractions and monitored by LC/MS. Fractions containing target compounds were combined and subjected to 2D NMR (*dqf*-COSY) measurements and semi-preparative HPLC purification (see below).

## Purification or enrichment of ascarosides from target SPE fractions using semi-preparative HPLC

Target SPE fractions were combined, filtered, dried, and finally dissolved in 280 µl MeOH, which was subjected to semi-preparative HPLC for further purification or enrichment (*Supplementary file 3—Table 6*). Before sample application, a mixture solvent (97% $H_2O$ and 3% ACN + 0.1% formic acid) was used to wash and equilibrate the HPLC system with a flow rate of 2.0 ml/min. Gradient washing steps with the application of 20 µl samples (injection volume for each run) were used by increasing the ACN content from 3% to 100% within the next 30 min, followed by 5 min isocratic washing using 100% ACN. Within the next 5 min, the ACN content in the solvent was decreased to 3% ACN, which lasted for another 10 min to equilibrate the HPLC system. Eluents were collected in 177 fractions and monitored by LC-MS. Fractions containing target compounds were combined and dissolved in $CD_3OD$ for acquisition of 1D and 2D NMR spectra.

## Phylogenetic analyses

Comparative analyses were conducted using a rescaled and pruned version of the *Pristionchus* phylogeny of *Rödelsperger et al., 2018* (*Figure 1—figure supplement 1*). This tree had been estimated via maximum likelihood analysis of a concatenated alignment of ~350,000 AAs. Here, the non-*Pristionchus* outgroups were dropped and the rooted ingroup was rescaled to generate an ultrametric tree using the penalized likelihood method of *Sanderson, 2002*, as implemented in the ape R package (*Paradis and Schliep, 2019*). For this, the optimal scaling factor (lambda) was determined via cross-validation using the `chronopl()` function (testing lambda = $10^k$ for k in −10 to 10), after which the tree was rescaled using the `chronos()` function with lambda = $10^6$ and a root age of 1. Species without NDMM profile data were then pruned, leaving a final ultrametric tree of 32 species.

We tested for phylogenetic signal in the qualitative distributions of NDMM compounds by estimating D statistics (*Fritz and Purvis, 2010*) using the `phylo.d()` function from the caper R package (*Orme et al., 2018*). This approach estimates the number of transitions between binary states (here, the presence or absence of the compound) across the phylogeny, standardized against the number of changes expected under Brownian evolution and under non-phylogenetic randomness. Here, departures from D = 1 are evidence of phylogenetic signal, with D < 1 suggesting increased conservation (with D = 0 being the special case of Brownian evolution) and D > 1 suggesting phylogenetic overdispersion. D cannot be estimated and tested for constant traits or for traits where a single species differs from all others. For each remaining compound, the null hypotheses D = 1 and D = 0 were evaluated by comparing observed D against expected D on the basis of N = 10,000 permutations, with the one-sided p-values provided by the `phylo.d()` function manually converted to two-sided p-values. P-values were adjusted to control the false discovery rate by applying the correction method of *Benjamini and Hochberg, 1995* and using an FDR threshold of 10% to gauge statistical significance; this was done separately for the two sets of tests. Note that estimation of D is sensitive to sample size (number of species) and trait prevalence (fraction of species possessing one state). *Fritz and Purvis, 2010* found that these tests had good power with sample sizes of 50, especially

when prevalence was moderate, but were somewhat underpowered for sample sizes of 25. Our study covers 32 species, suggesting that our results may be somewhat conservative.

To test for phylogenetic signal in a way that accounts for quantitative variation among NDMM profiles, we examined whether multivariate trait-space distances between NDMM profiles correlated with pairwise phylogenetic distances. Multivariate analyses were conducted using functions from the vegan (*Oksanen et al., 2019*) and ape (*Paradis and Schliep, 2019*) R packages. Here, the raw data table (i.e., the MS area-under-peak data for each NDMM) was closed and expressed as within-species percentages (i.e., as compositions). This was done to remove the potential effect of absolute-scale variation in NDMM production rates. Pairwise distances between NDMM profiles were estimated using the Bray-Curtis dissimilarity statistic (using the vegan package's `vegdist()` function). Pairwise phylogenetic distances between tips were extracted from the ultrametric *Pristionchus* phylogeny using the ape function `cophenetic()`. A Mantel Test was used to test for a relationship between the values in these two dissimilarity matrices, using the function `mantel()` and with the significance of the observed Spearman correlation coefficient evaluated using a null distribution generated via N = 999 permutations. The nature of the relationship between the two matrices was visualized via a scatterplot with a linear fit and a LOESS curve. Relationships among NDMM profiles were additionally visualized through non-metric multidimensional scaling (NMDS) analysis of the compositional data set, using the vegan package's `metaMDS()` function with the following options: `distance='bray'`, `k = 2`, `autotransform = F`, `expand = F`.

NDMM evolutionary histories were reconstructed using the SIMMAP approach of *Bollback, 2006*, as implemented in the phytools R package (*Revell, 2012*). This is a Bayesian reconstruction method in which discrete traits are repeatedly mapped onto a phylogeny in light of a stochastic state-change model. As with the above-described estimation of binary-trait phylogenetic signal, ancestral reconstruction was not applied to NDMMs found in all species (which are simply assumed to be present at all ancestral nodes) or to singletons (which are simply assumed to have evolved on the relevant tip branch). We also reconstructed three NDMM-class traits that combined data across individual NDMMs for the NPAR, UBAS, and PASC classes. The state-change model for each case was selected through likelihood ratio tests of the 'equal rates' (ER) and 'all rates differ' (ARD) models, using the `fitMk()` function and assuming an equal prior distribution for the root. The more complex ARD model was favored in 13 of 43 cases when considering individual NDMMs; the ER model was favored in each case when considering NDMM-classes. SIMMAP was implemented using the `make.simmap()` function, using 100 simulations per NDMM, an equal root-state prior probability distribution, and the state-change model chosen for the given NDMM. The 'mcmc' option was selected to account for uncertainty in rate-change estimates within the state-change matrix. Ancestral reconstructions were extracted using the `describe.simmap()` function, with a particular focus on the nodes that define the four major *Pristionchus* ingroup clades (i.e., the *pacificus*, *maupasi*, *entomophagus*, and *triformis* clades).

## Statistics

R version 3.3.2 (http://www.R-project.org/) was used to perform statistical analysis of the data. Statistical methods are indicated in the 'Phylogenetic analyses' section of the 'Materials and methods'.

## Acknowledgements

We thank Neelanjan Bose and Frank C Schroeder for the chemical analysis of exudates derived from *P. pacificus* males and hermaphrodites. We also thank Wen Hu for preparing and providing *P. pacificus* inbred strains for chemicals analysis. Synthetic standard chemical of ascr#1 was kindly provided by Stephan H von Reuss from the University of Neuchâtel. We are also grateful to Michael Reichelt (Department for Biochemistry, MPICE, Jena) for performing MS/MS precursor ion screening experiments, and to Matthias Herrmann and Christian Weiler for collecting *Pristionchus* nematodes. Transcriptome data for *Pristionchus* species provided by Christian Rödelsperger and discussions with Marc Claassen are gratefully acknowledged. CJW was supported by a Royal Society Dorothy Hodgkin Fellowship during the latter stages of this project. This work is funded by the Max Planck Society.

## Additional information

### Funding

| Funder | Grant reference number | Author |
| --- | --- | --- |
| Max-Planck-Gesellschaft | | Ralf J Sommer |
| Royal Society | Dorothy Hodgkin Fellowship | Cameron J Weadick |

The funders had no role in study design, data collection and interpretation, or the decision to submit the work for publication.

### Author contributions

Chuanfu Dong, Conceptualization, Data curation, Formal analysis, Investigation, Methodology, Writing - original draft; Cameron J Weadick, Vincent Truffault, Investigation, Methodology, Writing - review and editing; Ralf J Sommer, Conceptualization, Funding acquisition, Writing - original draft, Project administration, Writing - review and editing

### Author ORCIDs

Chuanfu Dong (iD) https://orcid.org/0000-0003-3043-7257
Cameron J Weadick (iD) https://orcid.org/0000-0001-8022-1783
Ralf J Sommer (iD) https://orcid.org/0000-0003-1503-7749

### Decision letter and Author response

Decision letter https://doi.org/10.7554/eLife.55687.sa1
Author response https://doi.org/10.7554/eLife.55687.sa2

## Additional files

### Supplementary files

• Supplementary file 1. NMR spectra of isolated or enriched NDMMs from *Pristionchus* species. a. NMR spectra of simple ASCR. b. NMR spectra of DASC chemicals. c. NMR spectra of PASC chemicals. d. NMR spectra of UBAS chemicals. e. NMR spectra of UPAS chemicals.

• Supplementary file 2. MS/MS spectral data of NDMMs detected from *Pristionchus* species. a. MS/MS spectral data of DASC chemicals. b. MS/MS spectral data of PASC chemicals. c. MS/MS spectral data of UBAS chemicals. d. MS/MS spectral data of UPAS chemicals.

• Supplementary file 3. Detected NDMMs from *Pristionchus* species and NMR data for isolated or enriched NDMMs from *Pristionchus* species.

• Supplementary file 4. Ancestral reconstruction for individual compounds.

• Transparent reporting form

### Data availability

All data generated during this study are included in the manuscript and supporting files. Source data files have been provided.

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
