## [Decision Letter]

**Acceptance summary:**

This data-rich comparison of the small molecule pheromone suites of two different nematodes – one a popular laboratory model – provides a detailed account of the functional convergence of genetically encoded signaling molecules. It also provides the basis for future work on these systems and a model for incorporating new systems.

**Decision letter after peer review:**

Thank you for submitting your article "Rapid evolution of small-molecule architecture in *Pristionchus*" for consideration by *eLife*. Your article has been reviewed by three peer reviewers, including Jon Clardy as the Reviewing Editor and Reviewer #1, and the evaluation has been overseen by Michael Marletta as the Senior Editor. The following individual involved in review of your submission has agreed to reveal their identity: Paul W Sternberg (Reviewer #3).

All three reviewers appreciated both the quantity and quality of the data reported in this manuscript along with its potential for informing our understanding of the evolution of small molecule signals. They also felt that there were shortcomings with current version of the manuscript that would require significant revisions but no additional experiments. In general the revisions suggested an effort to make the article more accessible and to discuss technological limitations and caveats to a greater extent.

In more detail:

The Introduction of the manuscript should be rewritten to place the study in a context accessible to the general reader. Beginning with why nematodes are studied and what we've learned from them would be very useful or why studying the evolution of small molecule signals is of broad interest.

This manuscript analyzes the ascarosides and paratosides present in several non-*Pristionchus* diplogastrids and a large number of *Pristionchus* species and uses this information to describe patterns seen in the evolution of different compound classes, such as the independent development of the same type of ascaroside in two separate clades. These patterns are interesting, and the manuscript might be presented better if the authors focused on these patterns. As written, the results begin with a number of findings related to simple ascarosides, many of which are not backed up by experimental data (such as the possible role of asc-C9 as a sex pheromone), and these results could be consolidated.

There are general statements that need additional qualification or explanation, like the use of rapid. Rapid compared to what? Also, the manuscript suggests that *P. pacificus* produces complex ascarosides but *C. elegans* does not, but it could easily be argued that both produce complex ascarosides. *C. elegans* produces ascarosides with a variety of head and terminus modifications, including ascarosides with multiple modifications and modifications that include multiple types of groups, and many of the modifications can be quite complex. The authors cite several papers to show that Caenorhabditis nematodes produce relatively simple ascarosides, but many of these papers (Choe et al., 2012, and Dong et al., 2018) used detection methods that could only detect previously identified and/or unmodified ascarosides. It would be better to focus on defining why and to what degree *Pristionchus* produces ascarosides with modifications that have not yet been found in other nematodes.

The findings of the manuscript are quite interesting in their own right, and that should be the focus, not greater complexity of faster evolution.

---

## [Author Response]

All three reviewers appreciated both the quantity and quality of the data reported in this manuscript along with its potential for informing our understanding of the evolution of small molecule signals. They also felt that there were shortcomings with current version of the manuscript that would require significant revisions but no additional experiments. In general the revisions suggested an effort to make the article more accessible and to discuss technological limitations and caveats to a greater extent.

We are thankful to this overall recommendation. We have tried throughout the manuscript to make the text more accessible to a general readership. This starts already with the Abstract and continues throughout the whole manuscript.

In more detail:The Introduction of the manuscript should be rewritten to place the study in a context accessible to the general reader. Beginning with why nematodes are studied and what we've learned from them would be very useful or why studying the evolution of small molecule signals is of broad interest.

We were very pleased that the reviewers and editors encouraged us to have a longer and more general Introduction. We have followed this advice and have added 3 additional paragraphs at the beginning of the Introduction that introduce small molecule/pheromone communication in general and the suitability of nematodes for such studies. We have moved one paragraph with more specific details of our study system to the beginning of the Result section to even further increase readability of the manuscript.

This manuscript analyzes the ascarosides and paratosides present in several non-Pristionchus diplogastrids and a large number of Pristionchus species and uses this information to describe patterns seen in the evolution of different compound classes, such as the independent development of the same type of ascaroside in two separate clades. These patterns are interesting, and the manuscript might be presented better if the authors focused on these patterns. As written, the results begin with a number of findings related to simple ascarosides, many of which are not backed up by experimental data (such as the possible role of asc-C9 as a sex pheromone), and these results could be consolidated.

We followed the suggestions to try to concentrate on the interesting evolutionary patterns including clade-specificities and convergent evolution of pheromonal small-molecule in *Pristionchus* nematodes. Specifically, we tried to describe the discovery of different classes of compounds in an evolutionary context (followed by structural characterization of novel compounds, as needed), starting with the simple ascarosides before describing compounds of greater degrees of complexity (dimeric ascarosides, and modular chemicals such as NPAR, PASC and UBAS). The paragraph on ascr#10 is now reduced and combined with results for other simple ascarosides. Next, we added one new paragraph to highlight the convergent evolution of small molecules given that we observed biosynthesis of PASC and UBAS compounds in both *pacificus*- and *triformis*-clades. Furthermore, we reorganized our presentation of the evolutionary analysis that clearly indicated convergent evolution for these identified chemicals, as well as our section an outgroup species, so as to improve emphasis and readability. Finally, we mentioned the chemical diversity and great biosynthetic capacity by focusing on some of those trace compounds. We believe that these arrangements make the text much more focused on the most interesting findings of clade-specificity and convergent evolution of small molecules in *Pristionchus* nematodes (changes are in blue in the marked copy of the manuscript).

There are general statements that need additional qualification or explanation, like the use of rapid. Rapid compared to what? Also the manuscript suggests that P. pacificus produces complex ascarosides but *C. elegans* does not, but it could easily be argued that both produce complex ascarosides. *C. elegans* produces ascarosides with a variety of head and terminus modifications, including ascarosides with multiple modifications and modifications that include multiple types of groups, and many of the modifications can be quite complex. The authors cite several papers to show that Caenorhabditis nematodes produce relatively simple ascarosides, but many of these papers (Choe et al., 2012, and Dong et al., 2018) used detection methods that could only detect previously identified and/or unmodified ascarosides. It would be better to focus on defining why and to what degree Pristionchus produces ascarosides with modifications that have not yet been found in other nematodes.

We have basically followed all of these suggestions:

1) We removed the term “rapid” from the title and Abstract, and from most locations throughout the Results and Discussion sections. Instead, we followed the recommendation to focus on the patterns of convergent evolution observed in our study. We have therefore, for example, changed our title introducing the convergent evolution aspect (see also comment below).

2) We have better defined the NDMM patterns of *P. pacificus* as requested.

3) Also, we specifically addressed the methodologies used in the various comparative studies on nematode NDMMs.

The findings of the manuscript are quite interesting in their own right, and that should be the focus, not greater complexity of faster evolution.

We are again thankful for this overall recommendation. In response, we have downplayed the complexity and faster evolution arguments altogether and focus more on the specific findings. This resulted not only in a change of title and Abstract (as already indicated above), but also re-organization of the Results and Discussion section as already indicated above.